# Opposing functions of F-BAR proteins in neuronal membrane protrusion, tubule formation, and neurite outgrowth

Kendra L Taylor[1], Russell J Taylor[1], Karl E Richters[2], Brandon Huynh[2], Justin Carrington[2], Maeve E McDermott[2], Rebecca L Wilson[2], Erik W Dent[2]

The F-BAR family of proteins play important roles in many cellular processes by regulating both membrane and actin dynamics. The CIP4 family of F-BAR proteins is widely recognized to function in endocytosis by elongating endocytosing vesicles. However, in primary cortical neurons, CIP4 concentrates at the tips of extending lamellipodia and filopodia and inhibits neurite outgrowth. Here, we report that the highly homologous CIP4 family member, FBP17, induces tubular structures in primary cortical neurons and results in precocious neurite formation. Through domain swapping and deletion experiments, we demonstrate that a novel polybasic region between the F-BAR and HR1 domains is required for membrane bending. Moreover, the presence of a poly-PxxP region in longer splice isoforms of CIP4 and FBP17 largely reverses the localization and function of these proteins. Thus, CIP4 and FBP17 function as an antagonistic pair to fine-tune membrane protrusion, endocytosis, and neurite formation during early neuronal development.

## Introduction

Membrane dynamics underlie many important biological processes in all cell types. Control of membrane protrusion and invagination and their effects on cell morphology requires coordination of both the plasma membrane and the actin cytoskeleton. The regulation of cell morphology is particularly important for the development of the brain. Cortical and hippocampal neurons undergo a series of stereotyped morphological changes as they develop into mature neurons (Kaech & Banker, 2006) After attachment to the substrate, neurons exhibit protrusive behavior by extending lamellipodia and filopodia (stage 1). Filopodial protrusions elongate into neurites, with actin-rich growth cones at their distal tips (stage 2). One neurite begins to extend rapidly to become the axon (stage 3), whereas the remaining neurites develop into dendrites (stage 4)

and form dendritic spines along their lengths (stage 5). These stages are readily apparent both in vitro and in vivo.

Although much is known about the processes responsible for axon formation and the latter stages of neuronal development (Namba et al, 2015; Bentley & Banker, 2016), mechanisms underlying the process of neuritogenesis have been less studied (Sainath & Gallo, 2015). Actin-driven filopodial and lamellipodial protrusion in early developing neurons control the essential process of neurite formation and require the coordination of the actin cytoskeleton and the plasma membrane (Dent et al, 2007; Gupton & Gertler, 2007; Flynn et al, 2012). The Bin–Amphiphysin–Rvs (BAR) domain proteins (including F-BAR, I-BAR, and N-BAR) have emerged as prominent players in linking the plasma membrane to actin dynamics in both endocytosis and protrusion (Salzer et al, 2017). BAR proteins form obligate dimers and assemble into polymeric complexes that allow them to bind and bend membranes. Thus, BAR proteins are likely to bridge the gap between actin polymerization and plasma membrane deformation and could play an important role in the regulation of neuritogenesis.

The F-BAR superfamily of proteins interact directly with negatively charged membrane phospholipids via an N-terminal F-BAR domain and are divided into several subfamilies based on the composition of the C-terminal end of the protein (Aspenstrom, 2009; Liu et al, 2015). Most F-BAR proteins are known to function in endocytosis. However, several members of the F-BAR superfamily can also induce membrane protrusions, including Slit-Robo GTPase–activating protein 2 (srGAP2), Cdc42-interacting protein 4 (CIP4), and nervous wreck (Nwk). These proteins have been shown to form filopodia (Guerrier et al, 2009), lamellipodia/veils (Saengsawang et al, 2012), and scallops/protrusions (Becalska et al, 2013) in various cell types, suggesting they could be classified as inverse F-BAR (iF-BAR) proteins. Moreover, these F-BAR proteins play important roles in neuronal development. SrGAP2 regulates leading process number and branching, and alterations in protein expression results in neuronal migration defects (Guerrier et al, 2009). CIP4 overexpression in early differentiating cortical neurons

[1]University of Wisconsin-Madison, Neuroscience Training Program, Madison, WI, USA [2]University of Wisconsin-Madison, Department of Neuroscience, Madison, WI, USA

Correspondence: ewdent@wisc.edu

produces rounded cells, with few filopodia, which results in the inhibition of neurite outgrowth, whereas CIP4 knockout neurons have precocious neurite outgrowth (Saengsawang et al, 2012). Nwk deletion results in a synaptic overgrowth phenotype at the larval neuromuscular junction in *Drosophila* (Coyle et al, 2004; O'Connor-Giles et al, 2008; Rodal et al, 2008).

Generally, F-BAR proteins function in either endocytosis or protrusion, but not in both processes. The F-BAR protein CIP4 functions in endocytosis and tubulates membrane in several cell lines (Itoh et al, 2005; Tsujita et al, 2006; Hu et al, 2009; Becalska et al, 2013); however, it localizes to the tips of protruding membrane structures in primary cortical neurons (Saengsawang et al, 2012; Saengsawang et al, 2013). We set out to determine the mechanism by which CIP4 could function in both tubulation and protrusion and how CIP4 function differed from FBP17, a close CIP4 family member.

# Results

### CIP4 and FBP17 act antagonistically in primary cortical neurons

There are three CIP4 family members: CIP4, formin-binding protein 17 (FBP17) and transducer of Cdc42-dependent actin assembly 1 (TOCA1) (Dawson et al, 2006). These proteins are highly homologous, consisting of an N-terminal F-BAR domain followed by an HR1 domain that binds active Rho GTPases and an SH3 domain that binds various actin-associated proteins and dynamin (Aspenstrom, 2009). Here, we focused on two of these family members, CIP4 and FBP17, which have been shown to be important in neuronal development and function. All CIP4 and FBP17 isoforms contain an N-terminal F-BAR/EFC domain, followed by HR1 and SH3 domains (Fig 1A). Short (S) and long (L) isoforms of CIP4 and FBP17 are produced through alternative splicing, with long isoforms containing an additional ~60 aa coded by their 9th and 10th exons, respectively (Fujita et al, 2002; Wang et al, 2002). Short isoforms have a truncated linker region (L1$_S$) between the F-BAR and HR1 domains (Fig 1A). All isoforms of CIP4 and FBP17 induce tubule formation in COS-7 cells (Fig 1C) (Tsujita et al, 2006; Bu et al, 2009). CIP4$_S$ is the only CIP4 isoform endogenously expressed in brain (Wang et al, 2002; Saengsawang et al, 2012), whereas FBP17$_L$ is the most abundantly expressed isoform in neural tissue (Fujita et al, 2002; Kakimoto et al, 2004).

Previously, we have shown that expression of CIP4$_S$ does not result in tubule formation in embryonic (E14.5) primary cortical neurons; rather, it is concentrated at the tips of protruding filopodia and lamellipodia/veils (Saengsawang et al, 2012; Saengsawang et al, 2013) (Figs 1B and S1A). Interestingly, expression of either CIP4$_L$ or FBP17$_L$ resulted in tubule formation in primary cortical neurons (Figs 1B and S1A), similar to those produced in COS-7 cells (Fig 1C). FBP17$_S$ was distributed throughout the cytoplasm, rather than at peripheral protrusions such as CIP4$_S$. This differential distribution of the four CIP4 and FBP17 isoforms resulted in profound changes in cell shape. We quantified both localization of the protein and cell shape changes by measuring four different parameters: peripheral intensity, filopodial number, complexity, and tubule number. Cell complexity is defined as the ratio of cell perimeter to cell area. CIP4$_S$

was highly concentrated at the cell periphery (Figs 1D and S1B). We could not directly compare the levels of overexpression of proteins via Western blot because of the fact that only a relatively small proportion of cells are transfected in each preparation (20–40%). Moreover, all the antibodies we have tested label CIP4 knockout neurons, as we have documented previously (Saengsawang et al, 2012), making comparison on a cellular level unfeasible. For all of the experiments herein, we chose neurons that were expressing low to medium levels of the fluorescently labeled proteins, compared with other transfected cells in the dish, in an effort to limit any over-expression artifacts. CIP4$_S$ expressing neurons contained few filopodia (Figs 1E and S1D) or tubules (Figs 1G and S1E) and exhibited decreased cell complexity (Figs 1F and S1C). This morphology and distribution was significantly different than EGFP expression alone (Fig S1A–F). In contrast, CIP4$_L$ and FBP17$_L$ had very low peripheral intensity, more filopodia and tubules, and a higher complexity than CIP4$_S$ (Fig 1D–G). FBP17$_S$ expression resulted in an overall phenotype similar to EGFP expression, with a more diffuse localization within the neuron and an intermediate phenotype in regards to cell morphology (Fig 1D–G). Together, these results indicate that CIP4$_S$ and FBP17$_L$, the isoforms present in embryonic cortical neurons, have opposing distributions that result in markedly different morphology in stage 1 cortical neurons.

Previous work from our laboratory has shown that expression of CIP4$_S$ has profound effects on neuronal development, inhibiting neurite outgrowth and, therefore, retarding cell-stage progression from stage 1 to stage 2 (Saengsawang et al, 2012). Surprisingly, FBP17$_L$ expression resulted in the opposite phenotype, precocious neurite outgrowth and cell-stage progression (Fig 1H). The increase in complexity (Fig S1C) and filopodial length (Fig S1F) with FBP17$_L$ expression, relative to EGFP, may underlie the precocious neurite formation as filopodia are necessary for neurite initiation (Dent et al, 2007). Coexpression of both CIP4$_S$ and FBP17$_L$ at similar levels returned cell-stage progression to control (EGFP expression) levels (Fig 1H). These results suggest that CIP4$_S$ and FBP17$_L$ act antagonistically to regulate neurite outgrowth.

To better understand the molecular interaction between these closely related F-BAR proteins, we performed colocalization and coimmunoprecipitation (co-IP) experiments in primary neurons coexpressing different isoforms of CIP4 and FBP17. Because CIP4$_S$ and FBP17$_L$ are the major isoforms expressed in embryonic cortical neurons (Wakita et al, 2011; Saengsawang et al, 2012), we tested these first. Not surprisingly, coexpression of CIP4$_S$ and FBP17$_L$ resulted in little colocalization (Pearson's coefficient, R = 0.311) (Fig 1I and J). To further test this interaction, we coexpressed CIP4$_S$-HA with either GFP, FBP17$_L$-EGFP, or CIP4$_S$-EGFP (control) and immunoprecipitated using an anti-HA antibody. Consistent with the previous result, we found CIP4$_S$-HA did co-IP with CIP4$_S$-EGFP but not with FBP17$_L$-EGFP or GFP (Fig 1K and L). The lack of colocalization and co-IP suggests that CIP4$_S$ and FBP17$_L$ do not directly interact within developing neurons. However, CIP4$_L$ and FBP17$_L$ had a high degree of colocalization (R = 0.7509) and appeared to localize to the same membrane tubules in neurons (Fig S1G and H).

Unlike their localization in neurons, all CIP4 and FBP17 isoforms form tubules in COS-7 (Fig 1C) and HEK-293 cells (data not shown). F-BAR proteins form obligate dimers through their F-BAR/EFC domains, and previous studies have shown that CIP4 family

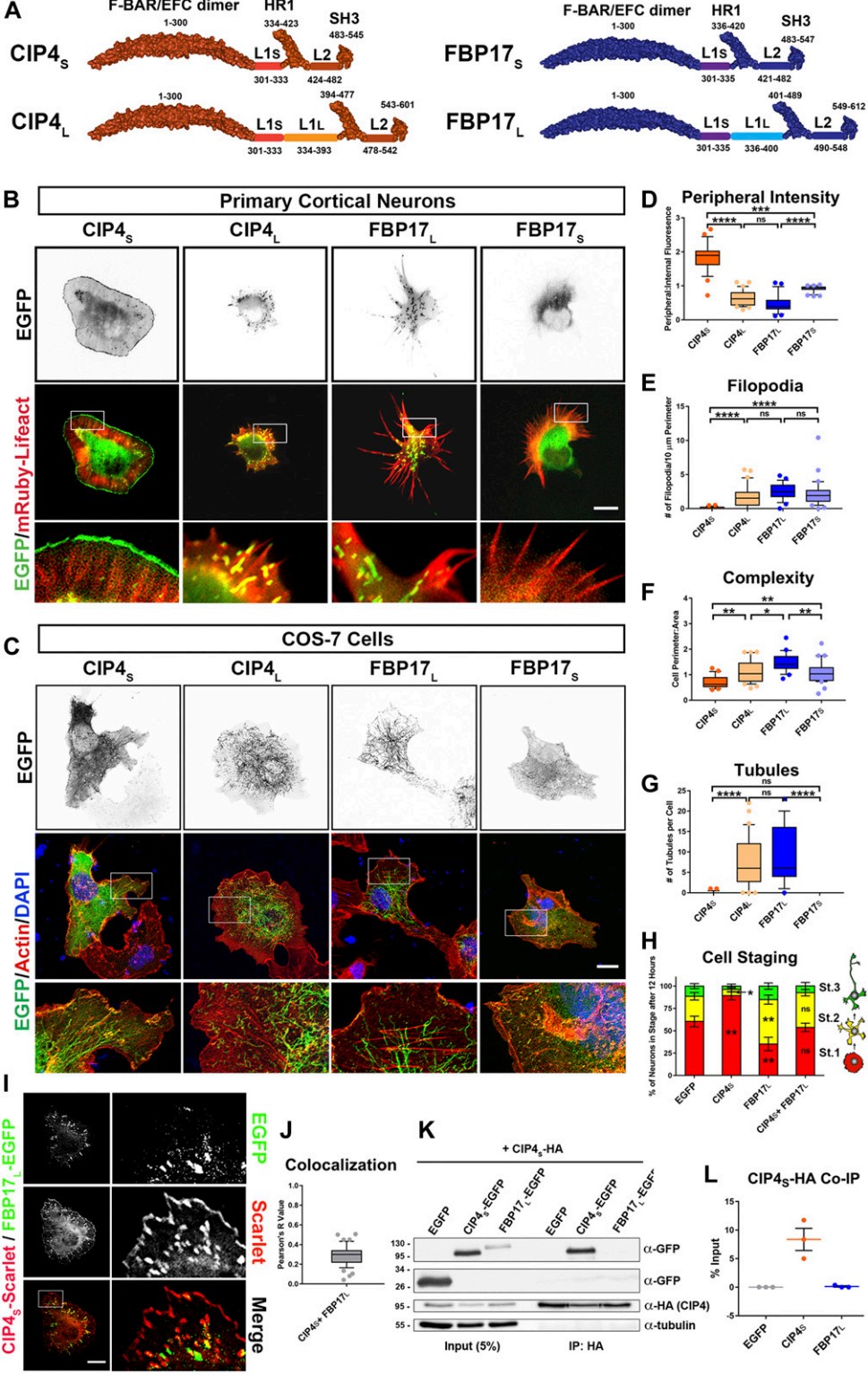

**Figure 1.  Long and short isoforms of CIP4 and FBP17 have opposing effects on cortical neuronal development.**
**(A)** Schematics of long and short human isoforms of CIP4 and FBP17. The F-BAR/EFC domain is shown as a dimer and only one C-terminal half of the protein is shown for clarity. F-BAR, HR1, and SH3 regions are false-colored, space-filling diagrams based on the following PDB files: CIP4 F-BAR/EFC domain (2EFK), FBP17 F-BAR/EFC domain (2EFL), HR1 domains (2KE4), and SH3 domains (2CT4). **(B)** Images of living cortical neurons at 12 h postplating, cotransfected with mRuby-Lifeact (red) to label actin and EGFP-labeled F-BAR protein (green). Contrast on black and white images is inverted for clarity. **(C)** Images of fixed COS-7 cells transfected with different isoforms of CIP4 and FBP17 and labeled with phalloidin (f-actin) and DAPI (nuclei). **(D–G)** Box-and-whisker plots showing quantification of stage 1 neurons (with points showing data that falls outside of the 10–90 percentile) comparing the effects of the different isoforms on peripheral intensity (D), filopodia number (E), cell complexity (F), and tubule number (G). CIP4$_S$-EGFP (n = 24 cells), CIP4$_L$-EGFP (n = 30 cells), FBP17$_L$-EGFP (n = 23 cells), or FBP17$_S$-EGFP (n = 31 cells). **(H)** Stacked bar graph comparing the percentage of neurons in stage (st.) 1, 2, and 3 for neurons expressing EGFP (n = 58), CIP4$_S$-EGFP (n = 72), FBP17$_L$-EGFP (n = 75), or CIP4$_S$-tdTomato and FBP17$_L$-EGFP (n = 65) at 12 h postplating. Two-way ANOVA with Bonferroni post-test multiple comparison. **(I)** Image of a living cortical neuron cotransfected with CIP4$_S$-Scarlet and FBP17$_L$-EGFP. **(J)** Box-and-whisker plot showing average colocalization (Pearson's correlation coefficient) of CIP4$_S$ and FBP17$_L$ in cortical neurons (n = 46 cells). **(K)** Co-IP with CIP4$_S$-HA and either CIP4$_S$-EGFP or FBP17$_L$-EGFP in cortical neurons. Original blot was separated to show higher molecular weight proteins (CIP4$_S$-EGFP and FBP17$_L$-EGFP) and EGFP. This blot was reprobed with antibodies to HA and tubulin. **(L)** Quantification of three co-IPs with CIP4$_S$-HA. One-way ANOVA with Kruskal–Wallis post-test multiple comparisons. *P < 0.05, **P < 0.01, ***P < 0.001, and ****P < 0.0001; ns, not significant. Scale bars represent 5 μm in whole-cell images of neurons and 1 μm in insets; 15 μm in whole-cell images of COS-7 cells and 7 μm in insets. Source data are available for this figure.

proteins must be able to both dimerize and multimerize through their F-BAR domains to bind and deform membrane (Frost et al, 2008; Shimada et al, 2010). Whereas F-BAR proteins are known to homodimerize, it is unknown whether CIP4 family members can

heterodimerize. As expected, in HEK-293 cells, CIP4$_S$ could co-IP with CIP4$_L$ and FBP17$_L$ could co-IP with FBP17$_S$ (Fig S1I and J). Both of these interactions were abrogated by deleting the F-BAR/EFC domain of CIP4$_S$ and FBP17$_L$ (Fig S1I and J), confirming that these

interactions were dependent on their respective F-BAR/EFC domains. However, we detected very little co-IP of CIP4$_S$ with either FBP17$_S$ or FBP17$_L$ (Fig S1I) and conversely little co-IP of FBP17$_L$ with either CIP4$_S$ or CIP4$_L$ (Fig S1J). These results indicate that CIP4 and FBP17 do not appear to form heterodimeric complexes. While there is colocalization of CIP4$_L$ and FBP17$_L$ in cortical neurons, it is likely due to both proteins binding to the same tubule, rather than directly forming heterodimers, as srGAP proteins are known to do (Coutinho-Budd et al, 2012). All of our data indicate that full-length CIP4 family proteins appear to form homodimers in neurons and HEK-293 cells and exert their effects on neuronal development independent of direct interaction with one another.

Because of this lack of heterodimerization, we were able to use chimeric swapping experiments to determine the function of the different domains of each protein. Moreover, because the studies throughout this article rely on the overexpression of proteins, we wanted to determine if fluorescently tagged CIP4$_S$ and FBP17$_L$ required endogenous CIP4 or FBP17 to localize to the periphery or tubules, respectively. To this end, we expressed CIP4$_S$-EGFP or FBP17$_L$-EGFP in cortical neurons from CIP4 knockout mice (Saengsawang et al, 2012). CIP4$_S$-EGFP clearly localized to the periphery (Fig S1K) in CIP4 knockout neurons and was indistinguishable from CIP4$_S$-EGFP expression in wild-type neurons (Figs 1B and S1A). FBP17$_L$ localized to tubules in CIP4 knockout neurons, as in wild-type neurons (Fig S1K). Unfortunately, shRNA to FBP17 did not knockdown endogenous expression within 12–24 h after plating primary cortical neurons (data not shown), which is the stage at which all analysis was conducted. This lack of knockdown precluded testing the localization of FBP17$_L$ in either an FBP17 knockdown or CIP4 knockout/FBP17 knockdown background. Therefore, most subsequent experiments were conducted in wild-type neurons.

### F-BAR, HR1, and SH3 domain swaps have little effect on protein localization and function

To determine the structural mechanism behind the dramatically different distribution and function of CIP4$_S$ and FBP17$_L$, we constructed chimeric proteins by swapping F-BAR, HR1, or SH3 domains. CIP4 and FBP17 contain the same five regions (Fig 1A). Because the L1 can be different lengths (CIP4$_S$ compared with CIP4$_L$—see Fig 1A), the letter that stands for L1 will have a subscript "S" or "L" to designate whether it is the short or long isoform, that is, C$_S$ or F$_L$ (Fig 2A). In an effort to clarify and summarize the results of subsequent manipulations of CIP4 and FBP17, all proteins, chimeras, and point and deletion mutations used in this study are shown in Table 1.

Because F-BAR proteins are subdivided into families based on the composition of the C-terminal end, we reasoned that the difference in CIP4$_S$ and FBP17$_L$ localization and function resided in the C-terminal half of the protein. Indeed, swapping the C-terminal half of the two proteins, resulting in CF$_L$FFF and FC$_S$CCC, demonstrated that the C-terminal half of CIP4$_S$ and FBP17$_L$ entirely controls their peripheral localization, complexity, number of filopodia, and number of tubules (Fig 2B–F). When compared with CIP4$_S$, the CF$_L$FFF chimera exhibited a dramatic decrease in peripheral intensity (Fig 2C) and an increase in filopodia (Fig 2D), complexity (Fig 2E), and tubules (Fig 2F). Likewise, FC$_S$CCC is significantly different from FBP17$_L$ in all metrics (Fig 2C–F). Interestingly, when CF$_L$FFF is compared with

FBP17$_L$ and FC$_S$CCC is compared with CIP4$_S$ (such comparisons would be considered F-BAR/EFC domain swaps), they are not significantly different in any of the four metrics (Fig 2C–F). Moreover, comparing CIP4$_S$ with CF$_L$FFF or FBP17$_L$ with FC$_S$CCC with regard to cell staging shows that the C-terminal half of the protein dictates the effect on neurite outgrowth (Fig 2G). CF$_L$FFF rescues the delay in cell staging progression observed with CIP4$_S$ expression and FC$_S$CCC reverses precocious neurite outgrowth and cell-stage progression observed with FBP17$_L$ expression (Fig 2G). These comparisons demonstrate that F-BAR domain swaps have no effect on CIP4$_S$ and FBP17$_L$ distribution and function. Rather, the differences between these two proteins are encoded in the C-terminal half of the protein.

To determine which domain(s) in the C-terminal half of the protein affect the localization and function of CIP4$_S$ and FBP17$_L$, we swapped the SH3 or HR1 domains between these two proteins. SH3 swap chimeras CC$_S$CCF and FF$_L$FFC had no effect on localization, filopodia number, or cell shape (Fig S2A–E) and only a minor effect on tubules (Fig S2F). Moreover, these chimeras did not result in any change in cell staging (Fig S2G). Surprisingly, this suggests that the different sets of proteins that are known to interact with the SH3 domain of CIP4$_S$ or FBP17$_L$, such as actin-associated proteins or dynamin, do not affect localization of the protein, cellular morphology, or progression of cell staging of cortical neurons.

The HR1 swap chimeras CC$_S$FCC and FF$_L$CFF (Fig 2H) had more complex effects. When the CIP4 HR1 domain was swapped into FBP17$_L$ (FF$_L$CFF), there was no effect on any of the four parameters we measured (Fig 2I–M) or in cell staging (Fig S2H). However, when the FBP17 HR1 domain was swapped into CIP4$_S$ (CC$_S$FCC), it resulted in decreased peripheral intensity and increased complexity and filopodia, but it did not result in tubule formation (Fig 2I–M). Consistently, these changes in morphology resulted in fewer neurons in stage 1 (Fig S2H). These results indicate that the CIP4 HR1 domain is responsible for a portion of the peripheral localization and resulting protruding lamellipodia/veils observed with CIP4$_S$ expression, but it is not sufficient to induce changes in localization or morphological changes induced by FBP17$_L$ expression. Furthermore, the CIP4 HR1 domain can substitute for the FBP17 HR1 domain in tubule formation. Surprisingly, the culmination of these domain swaps indicates that the three domains of CIP4$_S$ and FBP17$_L$ are largely interchangeable and are not responsible for the localization or function of these proteins in early developing neurons.

Nevertheless, the difference in peripheral intensity and filopodia number when the FBP17 HR1 domain was swapped into CIP4 suggests that the FBP17 and CIP4 HR1 domains are associating with different GTPases. In our previous study, we showed that in cortical neurons, CIP4 was sensitive to the activity of Rac1 rather than Cdc42 (Saengsawang et al, 2013). Because FBP17$_L$ localizes to tubules, we sought to determine if Cdc42 played a role in this localization. We expressed either constitutively active (CA) Cdc42-V12 or dominant negative (DN) Cdc42-N17 in neurons expressing FBP17$_L$ and discovered that DN-Cdc42 markedly decreased the number of FBP17-labeled tubules (Fig S3A and B). Moreover, we incubated FBP17-expressing neurons with ZCL278, a selective inhibitor of Cdc42 (Friesland et al, 2013) and discovered it decreased the number of tubules (Fig S3C and D). However, incubation with NSC23766, a selective Rac1 inhibitor (Gao et al, 2004) had no effect on the number of FBP17-containing tubules (Fig S3E). These data suggest

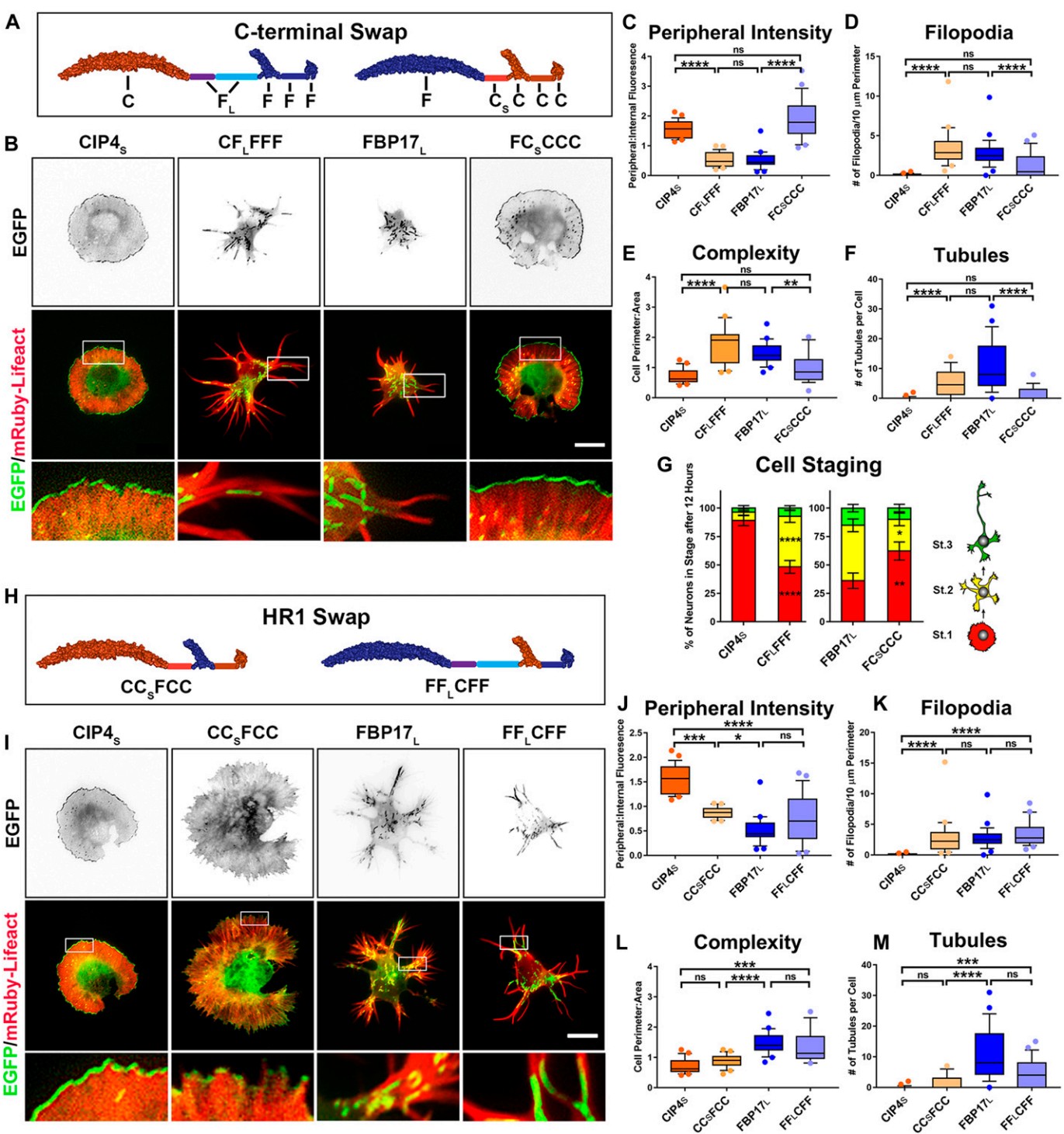

**Figure 2. The localization and function of CIP4$_S$ and FBP17$_L$ is encoded by the C-terminal half of the protein, but not by the HR1 domain alone.**
**(A)** Schematic of the C-terminal domain swaps CF$_L$FFF and FC$_S$CCC. Each letter represents a domain or region of the protein. **(B)** Images of living cortical neurons at 12 h postplating, cotransfected with mRuby-Lifeact, and EGFP-labeled protein or chimera. **(C–F)** Quantification of stage 1 neurons comparing the effects of the C-terminal swap constructs on peripheral intensity (C), filopodia number (D), cell complexity (E), and tubule number (F). CIP4$_S$-EGFP (n = 24 cells), CF$_L$FFF-EGFP (n = 22 cells), FBP17$_L$-EGFP (n = 24 cells), or FC$_S$CCC-EGFP (n = 21 cells). **(G)** Stacked bar graph comparing the percentage of neurons in stage (st) 1, 2, and 3 for neurons expressing CIP4$_S$-EGFP (n = 40) versus CF$_L$FFF-EGFP (n = 45) and FBP17$_L$-EGFP (n = 48) versus FC$_S$CCC-EGFP (n = 41). Two-way ANOVA with Bonferroni post-test multiple comparison. **(H)** Schematic of the HR1 domain swaps CC$_S$FCC and FF$_L$CFF. **(I)** Images of living cortical neurons cotransfected with mRuby-Lifeact and EGFP-labeled protein or chimera. **(J–M)** Graphs showing quantification of stage 1 neurons comparing the effects of the HR1 domain swap constructs on peripheral intensity (J), filopodia number (K), cell complexity (L), and tubule number (M). CIP4$_S$-EGFP (n = 24 cells), CC$_S$FCC-EGFP (n = 22 cells), FBP17$_L$-EGFP (n = 24 cells), or FF$_L$CFF-EGFP (n = 23 cells). One-way ANOVA with Kruskal–Wallis post-test multiple comparisons. *$P < 0.05$, **$P < 0.01$, ***$P < 0.001$, and ****$P < 0.0001$; ns, not significant. Scale bars represent 5 µm in whole-cell images and 1 µm in insets.

**Table 1.  Summary of all proteins, chimeras, and point and deletion mutants used in this study.**

| Protein/chimera/mutant | Nomenclature | Localization | Filopodia number | Tubule number | % Neurons in stage 1 |
|---|---|---|---|---|---|
| CIP4$_S$ | CC$_S$CCC | P | <1 | <1 | 90 |
| CIP4$_S$ SH3 swap | CC$_S$CCF | P | <1 | 1–2 | 90 |
| CIP4$_S$ + FBP17 L1$_S$ swap | CF$_S$CCC | P | <1 | <1 | ND |
| FBP17 + CIP4 L1$_S$/HR1/L2 swap | FC$_S$CCF | P | <1 | <1 | 85 |
| FBP17 C-terminal swap | FC$_S$CCC | P | 1–2 | 1–2 | 60 |
| CIP4$_S$ ΔL2+SH3 | CC$_S$C-- | P | 1–2 | 1–2 | 55[a] |
| CIP4$_S$ ΔSH3 | CC$_S$CC- | P | 0–1 | 1–2 | 55[a] |
| FBP17$_S$ + CIP4 HR1 swap | FF$_S$CFF | P | 1–2 | 1–2 | ND |
| CIP4$_L$ PxxP region mutant | CIP4$_L$-AxxA | P | 1–2 | 2–4 | ND |
| FBP17$_S$ ΔL2+SH3 | FF$_S$C-- | P | ND | ND | ND |
| FBP17 L1 swap | FC$_S$FFF | P | 1–2 | <1 | 65 |
| CIP4$_S$ HR1 swap | CC$_S$FCC | P/C | 2–4 | <1 | 65 |
| GFP | ----- | C | 2–4 | <1 | 60 |
| FBP17$_S$ | FF$_S$FFF | C | 1–2 | <1 | ND |
| CIP4$_S$ PBR mutant | CIP4$_S$-7Q | C | 1–2 | <1 | ND |
| CIP4$_L$ PBR mutant | CIP4$_L$-7Q | C | 1–2 | <1 | ND |
| CIP4 ΔL1 | C-CCC | C | 1–2 | <1 | ND |
| CIP4 F-BAR/EFC (1–300) | C---- | C | 1–2 | <1 | ND |
| CIP4 F-BAR/EFC + L1$_{S(7Q)}$ | CC$_S$---7Q | C | ND | ND | ND |
| CIP4 F-BAR/EFC + L1$_{L\,(7Q)}$ | CC$_L$---7Q | C | ND | ND | ND |
| FBP17 ΔL1 | F-FFF | C | 1–2 | <1 | ND |
| FBP17 F-BAR/EFC (1–300) | F---- | C | 1–3 | <1 | ND |
| FBP17$_L$ PxxP region mutant | FBP17$_L$-AxxA | V/T | 1–2 | 1–3 | ND |
| FBP17 F-BAR/EFC + L1$_S$ | FF$_S$--- | T | ND | ND | ND |
| CIP4 F-BAR/EFC + L1$_S$ | CC$_S$--- | T | 2–3 | 3–6 | ND |
| CIP4$_L$ | CC$_L$CCC | T | 1–2 | 4–8 | ND |
| CIP4 F-BAR/EFC + L1$_L$ | CC$_L$--- | T | ND | ND | ND |
| FBP17 F-BAR/EFC + L1$_L$ | FF$_L$--- | T | 2–3 | 4–8 | ND |
| CIP4 C-terminal swap | CF$_L$FFF | T | 2–4 | 4–8 | 50 |
| CIP4 L1 swap | CF$_L$CCC | T | 2–4 | 6–12 | 45 |
| CIP4 + FBP17 L1$_L$/HR1/L2 swap | CF$_L$FFC | T | 1–3 | 6–12 | 40 |
| FBP17$_L$ SH3 swap | FF$_L$FFC | T | 2–4 | 3–6 | 40 |
| FBP17$_L$ HR1 swap | FF$_L$CFF | T | 2–4 | 4–8 | 45 |
| FBP17$_L$ | FF$_L$FFF | T | 2–4 | 6–12 | 35 |

C, cytosol; ND, not determined; P, periphery; T, tubule; V, vesicle. Shading depicts the most CIP4$_S$ (orange) to the most FBP17$_L$ (blue) phenotype.
Staging determined at 12 h in vitro, Filopodia number is expressed per 10 μm of cell perimeter, and Tubule number is expressed per cell.
[a]In this set of experiments, only 75% of CIP4$_S$ neurons were in stage 1.

that FBP17$_L$ associates with Cdc42 in neurons, whereas our previous studies showed CIP4 associates with Rac1 in neurons.

## The first linker region largely determines the localization and function of CIP4$_S$ and FBP17$_L$

Because swapping the F-BAR, HR1, and SH3 domains was not sufficient to significantly alter localization and function of CIP4$_S$ and FBP17$_L$ in neurons, we focused on the two linker regions (L1 and L2) within these proteins. L1$_S$ is 33–35 aa in length and L1$_L$ is 99 aa (an additional 64–66 aa) in length (Fig 1A) and accounts for much of the size difference between CIP4$_S$ (545 aa) and FBP17$_L$ (612 aa). First, we made L1+HR1+L2 swaps (CF$_L$FFC and FC$_S$CCF) (Fig S4A). These two chimeric proteins resulted in the localization, neuronal morphology, and staging consistent with the middle regions of the proteins, not their respective F-BAR and SH3 domains (Fig S4B–G). When the

L2 linker was removed from FBP17$_L$ (FF$_L$F-F) or CIP4$_S$ (CC$_S$C-C), there was no change in localization of the deletion mutant (Fig S5E), suggesting the L2 linker plays little role in the protein localization or function. Because the HR1 domain swap had relatively minor effects on CIP4$_S$ and FBP17$_L$ (Fig 2H–M), we focused on the L1 linker region of these two proteins.

Surprisingly, transfection of L1 swap chimeras (Fig 3A) almost completely changed the localization and the morphology of stage 1 neurons (Fig 3B). Both the CF$_L$CCC and FC$_S$FFF chimeras significantly changed all four measured parameters when compared with CIP4$_S$ and FBP17$_L$, respectively (Fig 3C–F). Indeed, CF$_L$CCC was entirely indistinguishable from FBP17$_L$, and FC$_S$FFF only differed from CIP4$_S$ in filopodia number and complexity. Importantly, CF$_L$CCC localized to tubules and FC$_S$FFF localized to the peripheral protruding membrane in both wild-type neurons (Fig 3B) and CIP4 knockout neurons (Fig S5B), indicating endogenous CIP4 is not required for tubule or peripheral localization. Consistently, the percentage of neurons that remained in stage 1 significantly decreased with CF$_L$CCC and significantly increased with FC$_S$FFF (Fig 3G). These results suggest that the first linker region (L1) is largely responsible for the localization, morphology, and cell stage progression induced by CIP4$_S$ and FBP17$_L$ expression.

If the first linker region (L1) is indeed key to CIP4 and FBP17 localization, morphology, and function, then deleting this region should result in profound changes in these two proteins. Indeed, deletion of the first linker region in CIP4 (C-CCC) (Fig 4A) resulted in a complete loss of localization to the peripheral membrane (Fig 4B and C), whereas deletion of the first linker region in FBP17 (F-FFF) (Fig 4A) resulted in a complete loss of tubule localization (Fig 4B and F). Moreover, deletion of the first linker region resulted in a GFP-like distribution and significantly changed the number of filopodia (Fig 4D) and the complexity (Fig 4E) of cortical neurons. We wondered if the first linker regions alone (C$_S$ and F$_L$) were sufficient to localize to peripheral membrane and tubules, respectively. However, expression of either C$_S$ and F$_L$ resulted in a weak, uniform membrane label (data not shown). These results indicate that the first linker regions (L1) are necessary but not sufficient to localize or induce morphological changes in developing neurons.

Within cells, F-BAR domains alone are generally thought to be capable of binding and bending membranes into tubular structures (Tsujita et al, 2006; Shimada et al, 2007; Frost et al, 2008). Surprisingly, we found that expression of the F-BAR/EFC (1–300 aa) domain of either CIP4 or FBP17 was not sufficient to localize these proteins to membranous structures in either neurons (Fig 4A–C, F) or COS-7 cells (Fig S5A). Expression of either F-BAR/EFC domain resulted in dispersed, cytoplasmic labeling, similar to EGFP. However, a construct consisting of the F-BAR/EFC domain containing

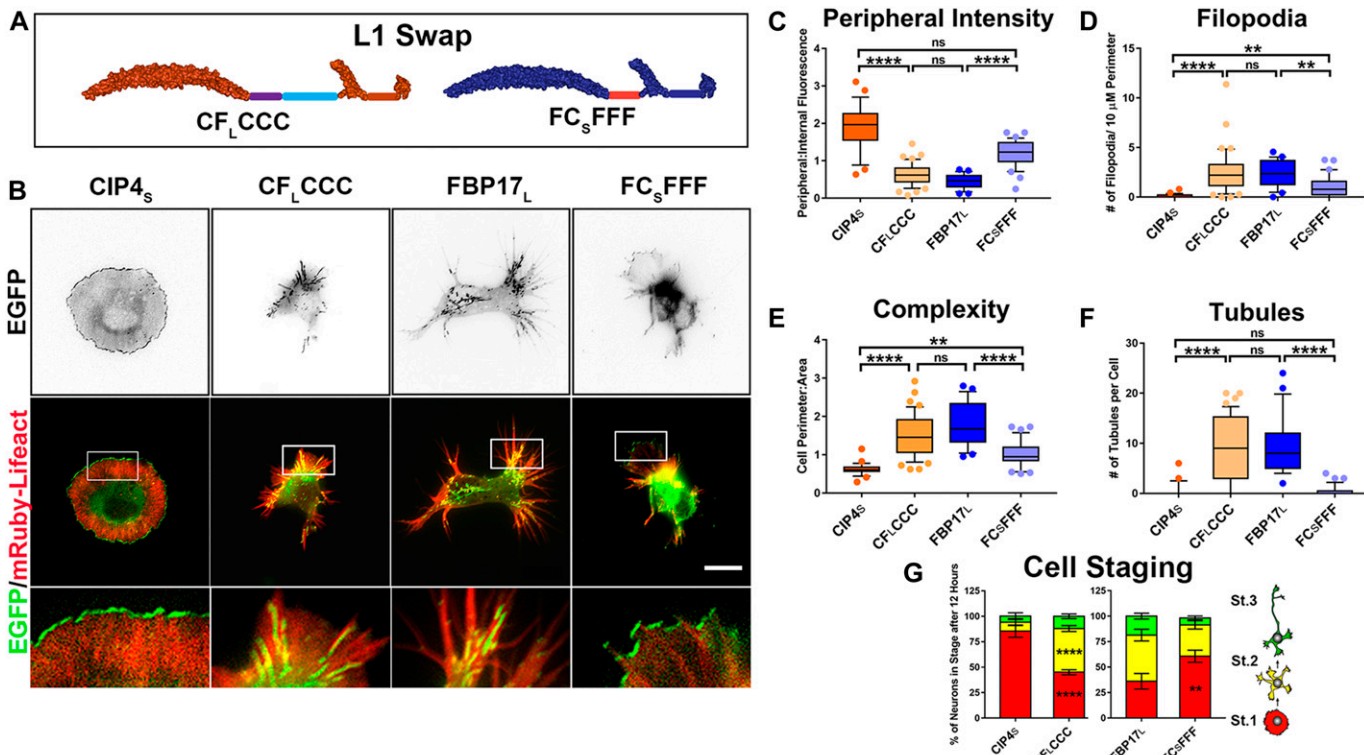

**Figure 3. Swapping the first linker region of CIP4$_S$ and FBP17$_L$ reverses localization and function.**
**(A)** Schematic of the L1 domain swaps CF$_L$CCC and FC$_S$FFF. **(B)** Images of living cortical neurons cotransfected with mRuby-Lifeact and EGFP-labeled protein or chimera at 12 h postplating. **(C–F)** Quantification of stage 1 neurons, comparing the effects of the L1 swap constructs on peripheral intensity (C), filopodia number (D), cell complexity (E), and tubule number (F) at 12 h postplating. CIP4$_S$-EGFP (n = 24 cells), CF$_L$CCC-EGFP (n = 47 cells), FBP17$_L$-EGFP (n = 23 cells), or FC$_S$FFF-EGFP (n = 37 cells). **(G)** Stacked bar graph comparing the percentage of neurons in stage (st) 1, 2, and 3 for neurons expressing CIP4$_S$-EGFP (n = 45) versus CF$_L$CCC-EGFP (n = 72) and FBP17$_L$-EGFP (n = 49) versus FC$_S$FFF-EGFP (n = 68) at 12 h postplating. Two-way ANOVA with Bonferroni post-test multiple comparison. *$P < 0.05$, **$P < 0.01$, ***$P < 0.001$, and ****$P < 0.0001$; ns, not significant. Scale bar represents 5 μm in whole-cell images and 1 μm in insets.

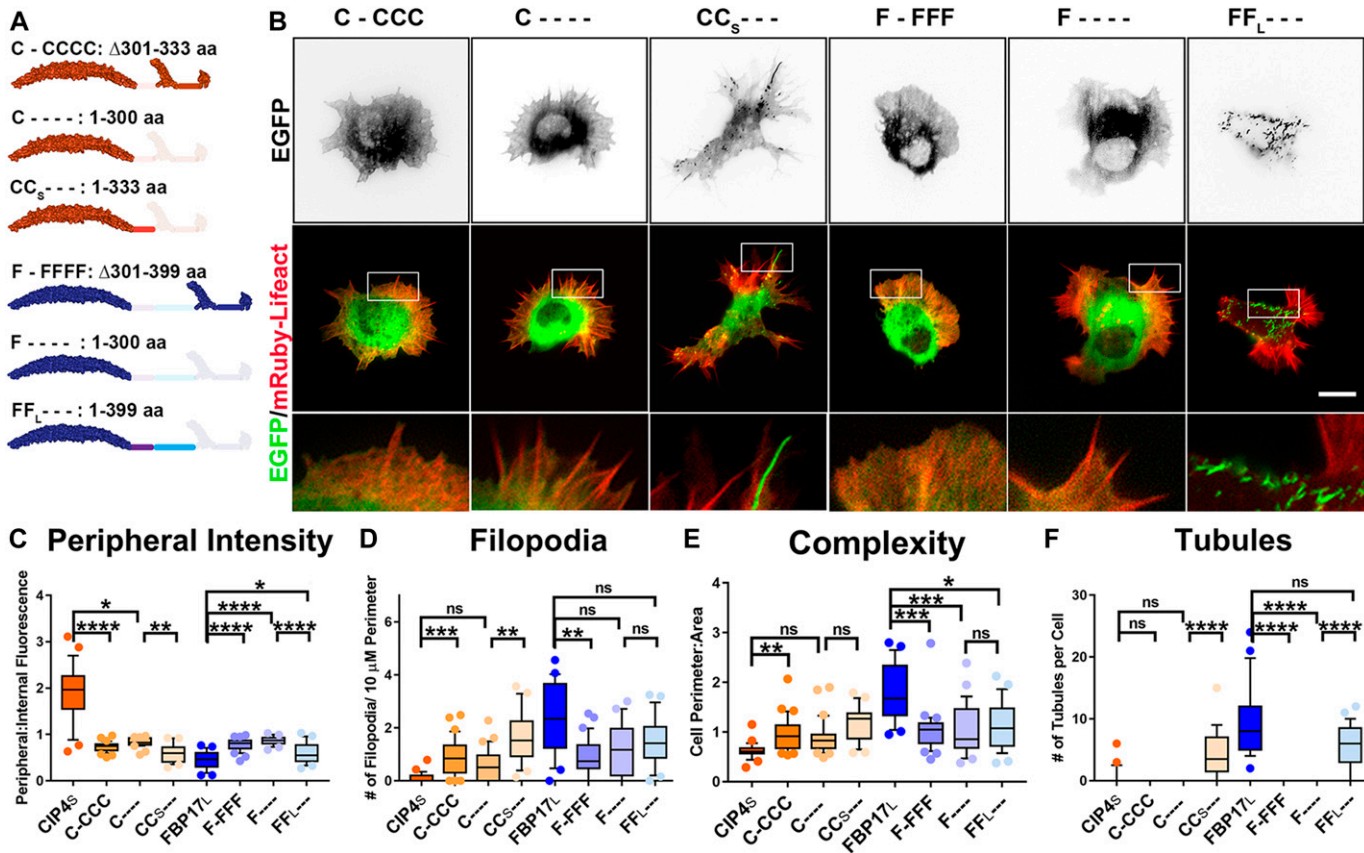

**Figure 4. The F-BAR and first linker region are required for membrane binding and bending.**
**(A)** Schematic of deletion constructs of CIP4$_S$ and FBP17$_L$. **(B)** Images of living cortical neurons cotransfected with mRuby-Lifeact and EGFP-labeled protein or deletion mutant at 12 h postplating. **(C–F)** Quantification of stage 1 neurons comparing the effects of the deletion constructs on peripheral intensity (C), filopodia number (D), cell complexity (E), and tubule number (F) at 12 h postplating. CIP4$_S$-EGFP (n = 24 cells), C-CCC-EGFP (n = 35 cells), C—— EGFP (n = 21 cells), CC$_S$— EGFP (n = 22 cells), FBP17$_L$-EGFP (n = 23 cells), F-FFF EGFP (n = 33 cells), F—— EGFP (n = 28 cells), and FF$_L$— EGFP (n = 29 cells). One-way ANOVA with Kruskal–Wallis post-test multiple comparisons. $*P < 0.05$, $**P < 0.01$, $***P < 0.001$, and $****P < 0.0001$; ns, not significant. Scale bars represent 5 μm in whole-cell images and 1 μm in insets.

the L1 region of either FBP17$_L$ (FF$_L$—) or CIP4$_S$ (CC$_S$—) was sufficient to localize to tubules but not peripheral membrane in neurons (Fig 4A–C and F). This was consistent with the localization of FBP17$_L$ to tubules but was unexpected for CC$_S$—, given that CIP4$_S$ concentrates strongly at peripheral protruding membranes. In fact, FF$_L$— and CC$_S$— were not significantly different from one another on any of the four measures (Fig 4C–F). Moreover, both CC$_L$— and FF$_S$— also localized to tubules in neurons (Fig S5D) and COS-7 cells (Fig S5C). Together, these results provide strong evidence that the first linker region of CIP4 and FBP17 (either L1$_S$ or L1$_L$) is necessary, and when coupled with the F-BAR/EFC domain, it is sufficient for tubule targeting and membrane deformation.

### Peripheral localization in neurons requires a short L1 linker and the CIP4 HR1 domain

To understand the molecular mechanism behind localization to protruding peripheral membrane in neurons, we created a chimeric protein where the HR1 domain of FBP17$_S$ was replaced by the HR1 domain of CIP4 (Fig 5A). The localization of this chimera, FF$_S$CFF, differed from FBP17s and mimicked CIP4$_S$ in all measures;

peripheral intensity, filopodia number, complexity, and tubules (Fig 5B–F). As further proof that the short linker and the CIP4 HR1 domain are necessary to bend membranes, we replaced the L1$_S$ in CIP4 with the L1$_S$ from FBP17 (Fig 5G). This chimera also mimicked CIP4$_S$ in all measures (Fig 5C–H). Thus, there is nothing inherently unique about the L1$_S$ region of CIP4$_S$, insofar as the L1$_S$ region of FBP17$_S$ can substitute for it.

Together, these data would suggest that, when coupled to an F-BAR/EFC domain (from either CIP4 or FBP17), a short linker region (from either CIP4 or FBP17) followed by the CIP4 HR1 domain is sufficient for peripheral localization and rounded cell morphology. To test this assertion, we transfected CC$_S$C– and FF$_S$C– and discovered that both constructs localized to the periphery and induced a rounded cell phenotype, whereas FF$_S$F– did not (Fig 5I–M). Thus, the HR1 domain of CIP4 is distinct from the HR1 domain in FBP17 and likely interacts with distinct GTPases, as we show above (Fig S3) and in previous work (Saengsawang et al, 2013). However, the deletion mutants CC$_S$C– and FF$_S$C– do not entirely mimic CIP4$_S$. They have significantly more filopodia, higher complexity (Fig 5K and L), and result in a decreased number and length of membrane protrusion events (Fig S6A–E). This suggests that although

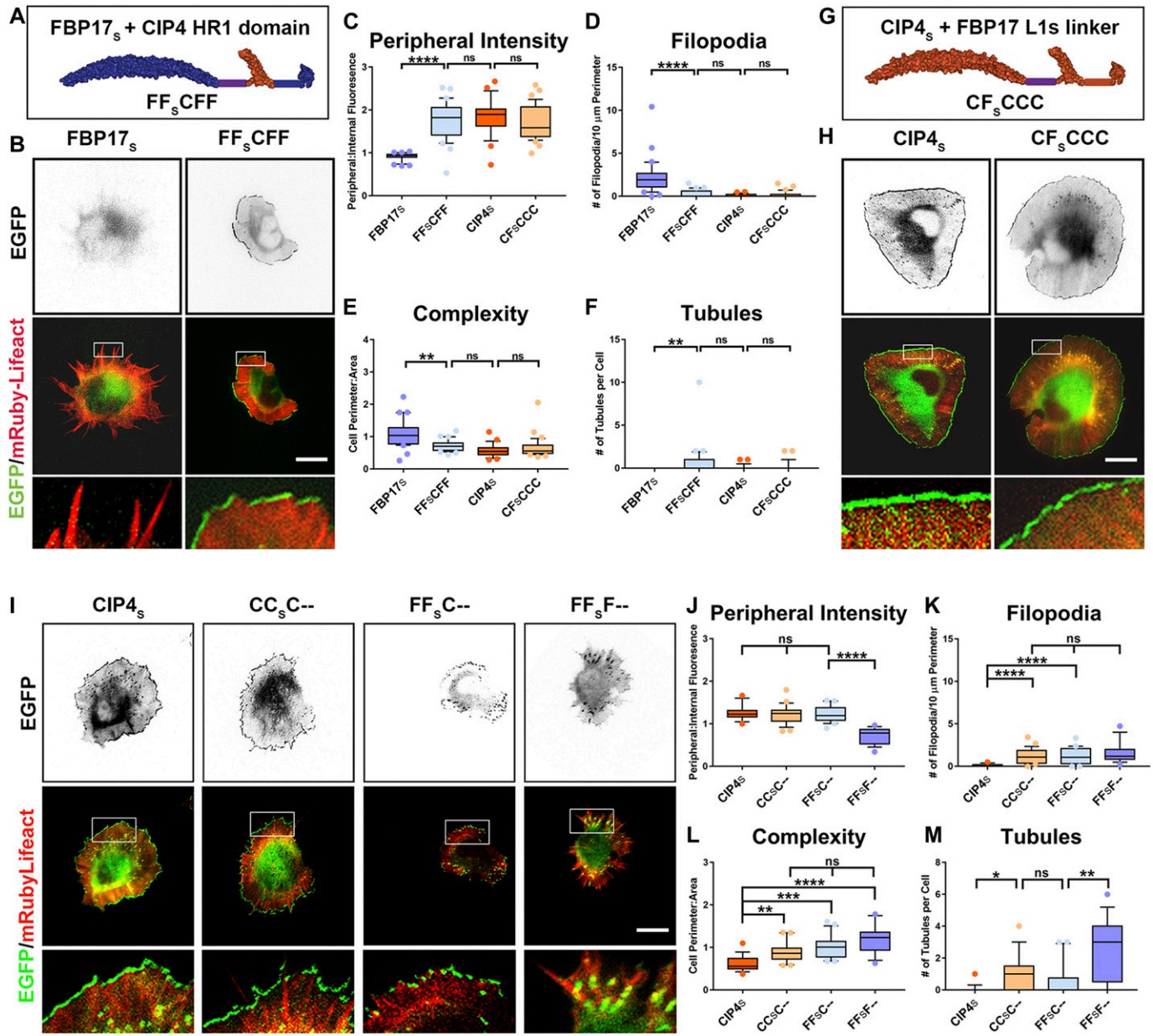

**Figure 5. The CIP4 HR1 domain is required for peripheral localization.**
**(A)** Schematic of the $FF_SCFF$ chimera. **(B)** Images of living cortical neurons cotransfected with mRuby-Lifeact and either EGFP-labeled protein or chimera at 12 h postplating. **(C–F)** Quantification of stage 1 neurons comparing the effects of chimeric constructs on peripheral intensity (C), filopodia number (D), cell complexity (E), and tubule number (F) at 12 h postplating. $FBP17_S$ (n = 31 cells), $FF_SCFF$ (n = 30 cells), $CIP4_S$ (n = 24 cells), and $CF_SCCC$ (n = 37 cells). **(G)** Schematic of the $CF_SCCC$ chimera. **(H)** Images of cortical neurons cotransfected with mRuby-Lifeact and either EGFP-labeled protein or chimera at 12 h postplating. **(I)** Images of cortical neurons cotransfected with mRuby-Lifeact and either EGFP-labeled protein or chimeric deletion at 12 h postplating. **(J–M)** Quantification of stage 1 neurons comparing the effects of chimeric deletion constructs on peripheral intensity (J), filopodia number (K), cell complexity (L), and tubule number (M) 12 h postplating. $CIP4_S$ (n = 16 cells), $CC_SC–$ (n = 25 cells), $FF_SC–$ (n = 20 cells), and $FF_SF–$ (n = 17 cells). One-way ANOVA with Kruskal–Wallis post-test multiple comparisons. $*P < 0.05$, $**P < 0.01$, $***P < 0.001$, and $****P < 0.0001$; ns, not significant. Scale bars represent 5 µm in whole-cell images and 1 µm in insets.

localization may be determined by $L1_S$ and the HR1 domain, neurite inhibition is dependent on the presence of an SH3 domain, likely through its interaction with actin-associated proteins. Consistently, deletion of just the SH3 domain ($CC_SCC–$) essentially replicates the localization and function of the $CC_SC–$ (Fig S6F–K), showing that the SH3 domain (and not L2) plays an important role in $CIP4_S$ function in neurons.

## The polybasic region (PBR) of $L1_S$ is necessary for membrane binding/bending

To determine the molecular mechanism by which the $L1_S$ region functions with the F-BAR/EFC domain for membrane binding and deformation, we examined the amino acid composition of this region. Upon close inspection of the CIP4 $L1_S$, we discovered a PBR

consisting of a stretch of positively charged lysine (K) and arginine (R) residues (between 315 and 328 aa), which are present in both short and long isoforms of FBP17 and CIP4 and are phylogenetically conserved (Fig S7A). To test whether the PBR is necessary for both $CIP4_S$ and $CIP4_L$ to localize to either peripheral membrane or tubules, respectively, we mutated seven positively charged lysine (K) and arginine (R) residues to neutral glutamine (Q) residues, resulting in $CIP4_S$-7Q and $CIP4_L$-7Q (Fig 6A). In both $CIP4_S$-7Q and $CIP4_L$-7Q, there is a complete loss of membrane localization (Fig 6B). $CIP4_S$-7Q shows a significant decrease in peripheral intensity (Fig 6B and C) and $CIP4_L$-7Q shows a matching decrease in the number of tubules (Fig 6B and F). As $CIP4_S$-7Q can no longer localize to the periphery, it is also no longer able to inhibit filopodia or induce lamellipodia/veil protrusion, resulting in increased filopodia and decreased complexity (Fig 6D and E). We also tested these constructs in COS-7 cells and discovered that neither $CIP4_S$-7Q nor $CIP4_L$-7Q could form tubules (Fig S7B). Together, these results show that the positive residues within the PBR are necessary for CIP4 to bind and deform the membrane in both primary cortical neurons and COS-7 cells.

However, the possibility exists that mutating seven positive residues could affect protein folding (Banerjee & Deniz, 2014) and potentially block the F-BAR domain from associating with membrane. To address this possibility, we made two additional mutant constructs, $CC_S$—7Q and a $CC_L$—7Q (Fig S7B and C). Because the CIP4 $L1_S$ and $L1_L$ are relatively small compared with the F-BAR domain, if the 7Q mutations affect protein folding, it is unlikely that the linkers would block the F-BAR domain. Expression of $CC_S$—7Q or $CC_L$—7Q resulted in a diffuse cytoplasmic distribution and a lack of tubule formation in both COS-7 cells (Fig S7B) and primary cortical neurons (Fig S7C). This localization is similar to that of the CIP4 F-BAR/EFC domain alone. Taken together, these results show that the PBR within the $L1_S$ of CIP4 is necessary for membrane localization, whether peripheral plasma membrane or tubules.

### The poly-proline (PxxP) region of $L1_L$ is necessary for tubule formation/elongation

Previous studies have shown that F-BAR proteins can be auto-inhibited by interactions between their F-BAR and SH3 domains, preventing the F-BAR from binding membrane (Rao et al, 2010; Stanishneva-Konovalova et al, 2016). By examining the $L1_L$ regions in CIP4 and FBP17, we discovered that both contain multiple PxxP motifs (Fig 6G). These PxxP motifs were confirmed as potential SH3 domain recognition sites when $CIP4_L$ and $FBP17_L$ protein sequences were entered into the Eukaryotic Linear Motif resource (elm.eu.org) (Dinkel et al, 2016). This suggests that the poly-PxxP region in the $L1_L$ of $CIP4_L$ and $FBP17_L$ may be capable of binding SH3 domains. Thus, an intra or intermolecular PxxP/SH3 interaction may provide a possible mechanism for the regulation of $FBP17_L$ and $CIP4_L$ in tubule formation.

To determine the role these PxxP motifs have in tubule formation, mutants were generated where all four of the PxxP motifs in $CIP4_L$ and all six of the PxxP motifs in $FBP17_L$ were mutated to alanines, resulting in AxxA motifs (Fig 6G). Interestingly, when the PxxP motifs are mutated, both $CIP4_L$-AxxA and $FBP17_L$-AxxA show localization and morphology similar to their respective short isoforms, $CIP4_S$ and $FBP17_S$ (Fig 6H). In particular, $CIP4_L$-AxxA shows an increased concentration at the peripheral membrane (Fig 6I) and significantly fewer tubules than $CIP4_L$ (Fig 6L). The effect of the AxxA mutation on $FBP17_L$ is less pronounced. $FBP17_L$-AxxA showed decreased complexity (Fig 6K) and tubule number (Fig 6L) but showed no significant changes in the peripheral intensity and filopodial number (Fig 6I and J). These data indicate that the poly-PxxP region in the long linker region of both $CIP4_L$ and $FBP17_L$ is necessary for normal tubule formation in cortical neurons. When these prolines are mutated to alanines, not only is there a marked reduction in tubule number, $CIP4_L$ and $FBP17_L$ start to take on characteristics of their short isoforms. Moreover, this poly-PxxP motif is conserved through most of phylogeny in both CIP4 and FBP17 (Fig S7A).

The data above suggest that $CIP4_L$ and $FBP17_L$ may be in an autoinhibited state, where the SH3 domain is potentially binding to the PxxP motifs and inhibiting the ability of the HR1 domain to associate with GTPases. We hypothesized that $CIP4_L$, which concentrates on tubules in neurons, might relocalize to the peripheral protruding membrane if it is activated. We attempted to activate $CIP4_L$ by coexpressing CA-Rac1 and discovered that Rac1 activation was sufficient to relocalize $CIP4_L$ to the peripheral protruding membrane and induce a rounded cell phenotype (Fig S8A), similar to $CIP4_S$ expression. However, CA-Rac1 expression had no effect on $FBP17_L$ (Fig S8A), consistent with our data that FBP17 is functioning with Cdc42 (Fig S3). Moreover, expression of CA-Rac1 in COS-7 cells did not relocalize $CIP4_L$ from tubules to the periphery (Fig S8B). These data show that although active Rac1 is necessary for peripheral localization of CIP4 in neurons, COS-7 cells may lack additional binding partners or the appropriate membrane composition to recruit CIP4 to the membrane.

## Discussion

Here, we demonstrate that the CIP4 family proteins $CIP4_S$ and $FBP17_L$ have opposing functions in early cortical neuron development by showing that $FBP17_L$ forms tubules and promotes precocious cortical neurite outgrowth, whereas $CIP4_S$ forms peripheral, protrusive veils and inhibits neurite outgrowth (Fig 7A). Mechanistically, we show that the opposing functions of these proteins and their disparate localization in cortical neurons are largely determined by two structural motifs in the intrinsically disorganized first linker region. The first motif is a PBR that is essential for CIP4 and FBP17 membrane binding and bending (Fig 7B). The second motif is a poly-PxxP region whose function is consistent with keeping the long isoforms of CIP4 and FBP17 directed to tubules, through autoinhibition of the C-terminal domain (Fig 7C). This "closed" configuration promotes tubule formation rather than protrusive veil formation. These findings describe, for the first time, the mechanism by which F-BAR proteins are able to differentially localize and function to either promote or inhibit neurite outgrowth, through tubule formation or protrusive veil formation, respectively.

### Opposing roles of CIP4 and FBP17 in neurite outgrowth

Formation of neurites is a fundamental process in neuronal development and required for subsequent axon and dendrite

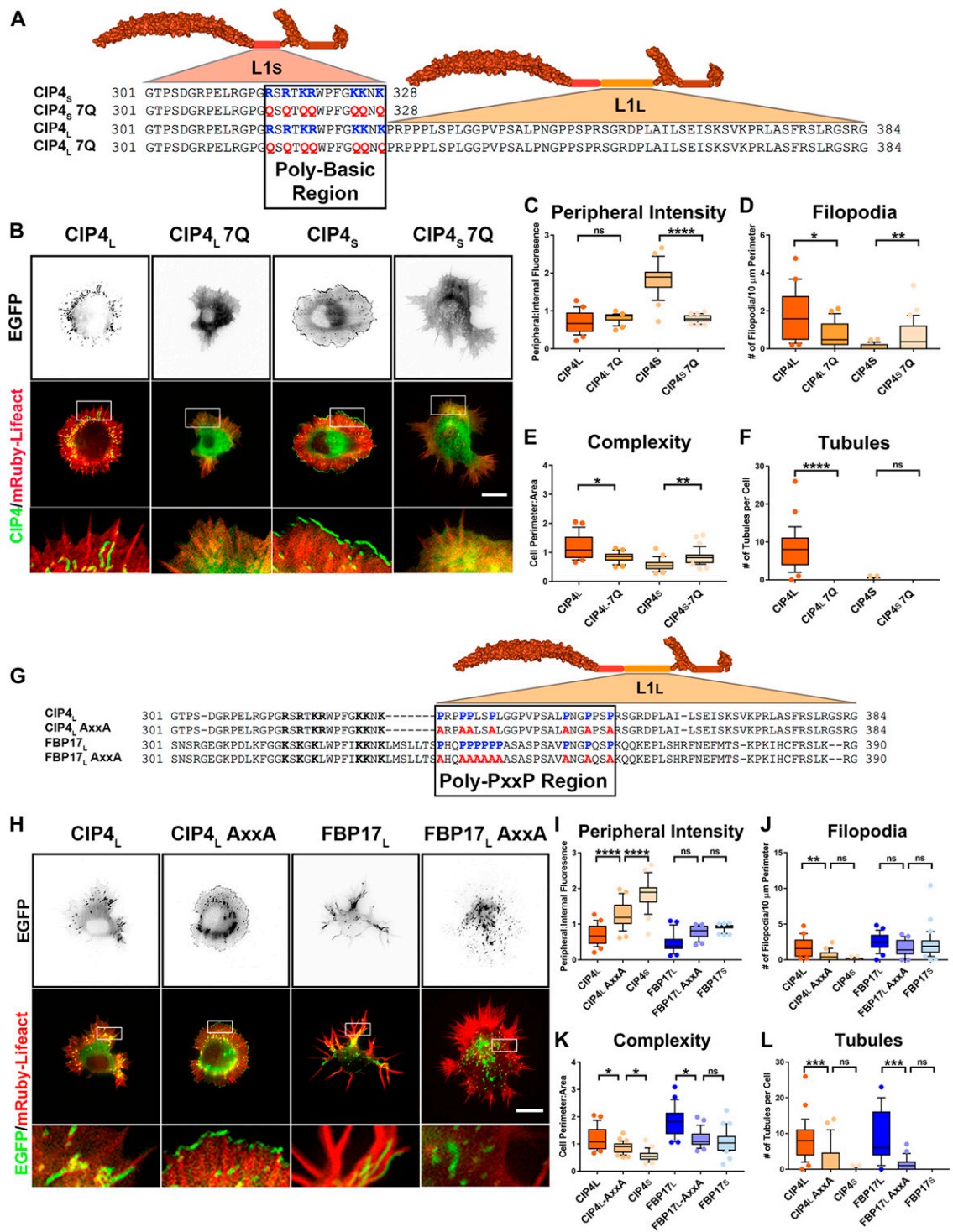

**Figure 6. The PBR is required for membrane bending and the poly-PxxP region is required for tubulation in cortical neurons.**
**(A)** Schematics of the CIP4 L1 PBR in the short and long isoforms, highlighting basic amino acids (K/R) in blue and the K/R-Q mutations in red. **(B)** Images of living cortical neurons cotransfected with mRuby-Lifeact and EGFP-labeled proteins or mutant proteins 12 h postplating. **(C–F)** Quantification of stage 1 neurons comparing the effects of the 7Q mutationson peripheral intensity (C), filopodia number (D), cell complexity (E), and tubule number (F) 12 h postplating. CIP4$_L$-EGFP (n = 29 cells), CIP4$_L$-7Q-EGFP (n = 28 cells), CIP4$_S$-EGFP( = 24 cells), or CIP4$_S$-7Q-EGFP (n = 34 cells). **(G)** Schematic of the L1$_L$ in CIP4$_L$ and FBP17$_L$ showing the PxxP motifs highlighted in blue and the AxxA mutations highlighted inred. **(H)** Images of living cortical neurons cotransfected with mRuby-Lifeact and either EGFP-labeled protein or mutant 12 h postplating. **(I–L)** Quantification of stage 1 neurons comparing the effects of the AxxA mutations on peripheral intensity (I), filopodia number (J), cell complexity (K), and tubule number (L) 12 h postplating. CIP4$_L$-EGFP (n = 29 cells), CIP4$_L$-AxxA-EGFP (n = 29 cells), CIP4$_S$-EGFP (n = 24 cells), FBP17$_L$-EGFP (n = 23 cells), FBP17$_L$-AxxA-EGFP (n = 25 cells), or FBP17$_S$-EGFP (n = 31 cells). One-way ANOVA with Kruskal–Wallis post-test multiple comparisons. *$P < 0.05$, **$P < 0.01$, ***$P < 0.001$, and ****$P < 0.0001$; ns, not significant. Scale bars represent 5 μm in whole-cell images and 1 μm ininsets.

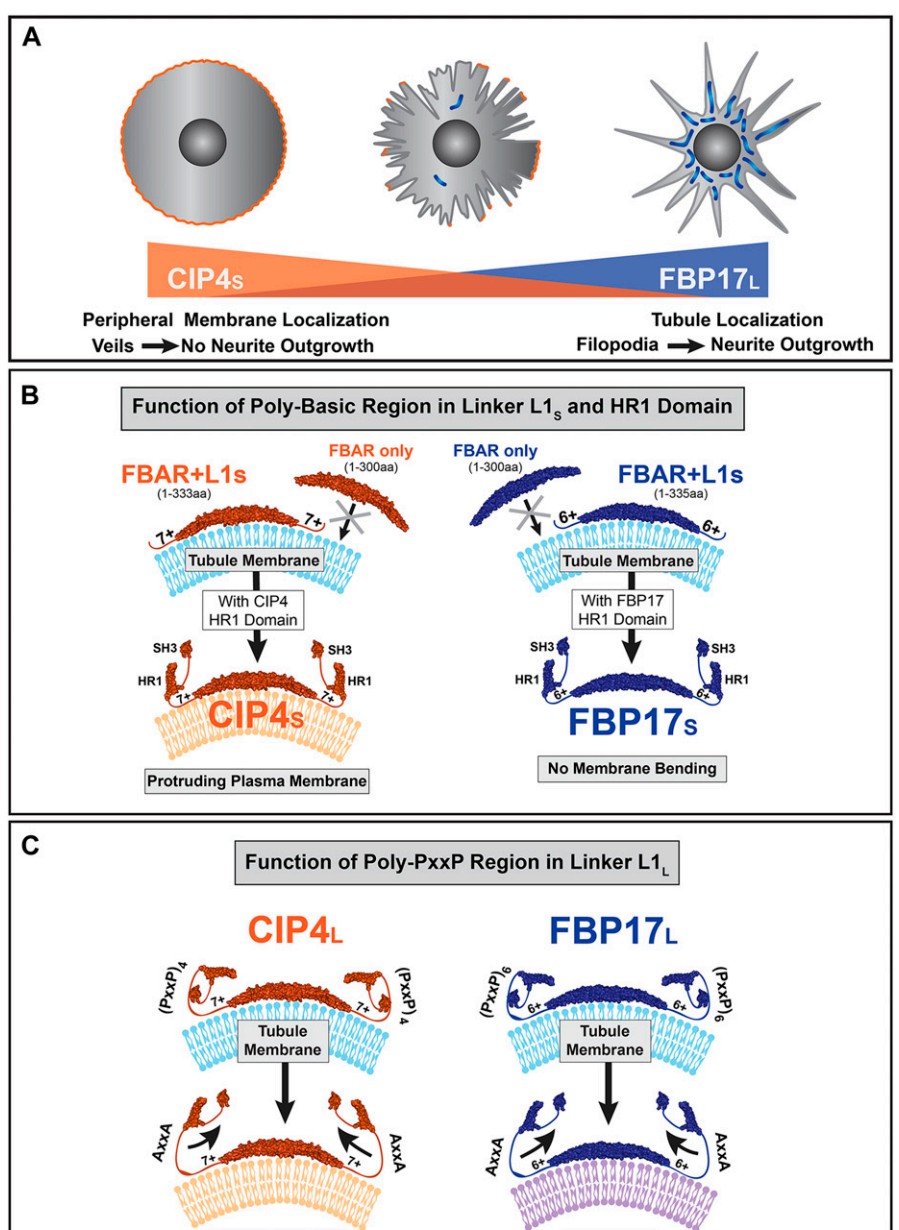

**Figure 7. Working model for CIP4 and FBP17 membrane localization and function.**
**(A)** Schematic showing the localization and function of $CIP4_S$ and $FBP17_L$. High levels of $CIP4_S$ expression results in concentration at the peripheral membrane, increased lamellipodia and veil protrusion, and inhibition of neurite outgrowth. In contrast, high levels of $FBP17_L$ expression results in concentration on tubules and excessive tubulation, more prominent filopodia formation, and promotion of precocious neurite outgrowth. A wild-type cell is shown with endogenous levels of $CIP4_S$ and $FBP17_L$ between these two extremes. **(B)** A model describing the function of the PBR and HR1 domain in $CIP4_S$ and $FBP17_S$. The F-BAR/EFC domain of CIP4 or FBP17 alone cannot bind or bend membrane. The F-BAR/EFC domains of CIP4 and FBP17 require the positive amino acid residues within the PBR to bind and tubulate membrane in primary cortical neurons (and COS-7 cells). The addition of the CIP4 HR1 domain relocates CIP4 to the peripheral plasma membrane, where it bends membrane but produces static (slowly extending/retracting) protrusions. Addition of the second linker region and SH3 domain results in dynamic (extending and retracting) protrusions. The addition of the FBP17 HR1 domain appears to prevent membrane bending, resulting in full-length $FBP17_S$ adopting a diffuse distribution, similar to EGFP. **(C)** A model describing the function of the poly-PxxP region in the $L1_L$ of $CIP4_L$ and $FBP17_L$. Long isoforms of both CIP4 and FBP17, which contain multiple PxxP motifs (four in $CIP4_L$ and six in $FBP17_L$), induce membrane tubulation in cortical neurons. When these poly-PxxP motifs are mutated to AxxA, the long isoforms are no longer able to form tubules and localize in a fashion similar to their short forms. As $CIP4_L$ contains the CIP4 HR1 domain, the AxxA mutation in $CIP4_L$ localizes to the peripheral membrane and causes lamellipodial/veil formation, whereas the AxxA mutation in $FBP17_L$ no longer tubulates, rather concentrating on vesicles.

formation (Tahirovic & Bradke, 2009). In addition, filopodial protrusion and neurite formation are intimately linked processes in early neuronal development (Dent et al, 2007). Here, we demonstrate a novel mechanism by which filopodia formation and subsequent neurite outgrowth can also be induced; through tubule formation by FBP17. Thus, we provide new evidence that invagination, resulting in tubule formation, is likely to work in concert with filopodial protrusion to promote neurite outgrowth. Nevertheless, to establish that tubule formation, prominent filopodia formation, and precocious neurite outgrowth are mechanistically linked, further work needs to be carried out to determine if tubule formation by other proteins is sufficient to induce precocious neurite outgrowth.

Since $CIP4_S$ and $FBP17_L$ do not colocalize or co-IP, as has been shown for srGAP family proteins (Coutinho-Budd et al, 2012), it stands to reason that CIP4 family members would have differing effects on neuronal development (Wakita et al, 2011; Saengsawang et al, 2012). A recent study, using only F-BAR domains (1–264 aa) of CIP4 and FBP17, showed these two protein domains co-IP with one another in RPE1 or HEK-293 cells and suggested that these proteins heterodimerize (Chan Wah Hak et al, 2018). However, a positive result through co-IP does not prove these are heterodimers, as the same result is possible with homodimeric proteins multimerizing with one another. Moreover, we show that the F-BAR/EFC regions (1–300 aa) of these two proteins, when

expressed in primary cortical neurons or COS-7 cells, exhibit a general cytoplasmic localization, similar to EGFP. Thus, the F-BAR domains do not show normal localization and may co-IP because they are both localized to the cytoplasm. We favor the interpretation that CIP4 and FBP17 do not normally associate as heterodimers.

Previous work has focused on the role of FBP17 in dendrite branching and spine formation (Fujita et al, 2002; Wakita et al, 2011), but this is the first study to examine FBP17 in early neuronal development. We show that FBP17$_L$ expression increases cell complexity and leads to early neurite formation. Moreover, we show that expression of FBP17$_L$ affects membrane remodeling and neurite development, likely through the processes of endocytosis and exocytosis (Tojima et al, 2010; Urbina et al, 2018). Further studies investigating the link between increased endocytosis and neurite initiation would clarify the role of FBP17$_L$-containing tubules in neuronal development. We favor the hypothesis that the timing of neurite formation in early development relies on the antagonistic roles CIP4 and FBP17 play in this process. We have shown previously that in developing cortex and in dissociated cortical neurons, CIP4$_S$ protein levels decrease throughout prenatal development, reaching almost undetectable levels soon after birth (Saengsawang et al, 2012), whereas FBP17$_L$ (otherwise known as Rapostilin L) protein levels are low prenatally and increase during early development (Wakita et al, 2011). Thus, we present a model where neurite outgrowth is initially inhibited by CIP4$_S$ (Fig 7A). As neurons mature, CIP4$_S$ levels decrease and FBP17$_L$ levels increase, favoring neurite outgrowth.

### The F-BAR domain and PBR are required for membrane bending

There have been several reports that the F-BAR/EFC domain of CIP4 family members are sufficient for tubulation in vitro (liposomes) (Shimada et al, 2007; Frost et al, 2008; Fricke et al, 2009) and in vivo (cells) (Kamioka et al, 2004; Tsujita et al, 2006; McDonald et al, 2015). However, F-BAR/EFC domains of differing length were used in these studies. We demonstrate that the F-BAR/EFC domains of CIP4 and FBP17 (1–300 aa) cannot induce membrane tubules in either COS-7 cells or primary cortical neurons. Our results are consistent with the studies of Kamioka et al (2004) and McDonald et al (2015), whose constructs were capable of forming long tubules when they contained the entire PBR (1–377) and short tubules with a truncated version of the PBR (1–319). Our data are also consistent with a study showing the F-BAR domain (1–331) of *Drosophila* CIP4/Toca-1 tubulated liposomes in vitro but could not induce membrane tubulation in *Drosophila* S2 cells (Fricke et al, 2009). As shown in Fig S7A, the amino acid sequence of *Drosophila* CIP4 (dCIP4) does not contain a poly-basic (or poly-PxxP) region and would, therefore, not be expected to be sufficient to induce membrane tubulation in cells. We suspect that cellular membrane has a more complex complement of lipids and proteins and may be harder to bind and bend than liposome membrane. Thus, having the polybasic region outside of the F-BAR domain would allow stronger association with the cellular membrane. It is unclear why our results are not consistent with the Tsujita study. We attempted to replicate these results by making an N-terminally tagged F-BAR/EFC domain of

CIP4 and FBP17, but these constructs also did not tubulate in COS-7 cells or neurons (data not shown). Thus, our results would suggest that CIP4 and FBP17 F-BAR/EFC domains (1–300 aa) should be classified with a growing number of other F-BAR/EFC domains, which are not capable of bending membranes in cells (McDonald et al, 2015) and require additional regions of the protein for this function.

The F-BAR/EFC domains of CIP4 and FBP17 are necessary for membrane interaction and are required to form both dimers and multimers as a structural basis for membrane tubulation (Shimada et al, 2007; Frost et al, 2008). In addition to the F-BAR/EFC domain, we show that for CIP4, a PBR is necessary, but not sufficient, for membrane deformation and localization to either membrane tubules or protrusive veils. A very similar stretch of positive amino acids is also present in the same L1$_S$ region of the other two CIP4 family members, FBP17 and TOCA1 (data not shown). This suggests that the PBR is important for all CIP4 family members to bind and bend membranes.

Recent work has identified other domains within the intrinsically disorganized regions of F-BAR family proteins, C-terminal to the F-BAR/EFC domain. These regions include the extended F-BAR domain (F-BARx) in FCHo2 (262–430 aa) (Henne et al, 2010), pacsin2/syndapin2 (304–369 aa) (Takeda et al, 2013), and srGAP2 (289–484 aa) (Sporny et al, 2016; Sporny et al, 2017), as well as the Fx/Fx(C) domains in FER (270–445 aa/415–434 aa) (Itoh et al, 2009; Yamamoto et al, 2018). All of these regions play significant roles in membrane binding and bending. The PBR that we have defined here in CIP4 family proteins differs from these regions and shares little homology with that of related F-BAR proteins, such as Nostrin, PSTPIP, FCHSD/Nwk, or the *S. pombe* F-BAR proteins Cdc15 or Imp2 (data not shown).

### The PxxP region may function in autoinhibition by binding the SH3 domain

Although it is well-known that F-BAR superfamily proteins are involved in membrane remodeling, the mechanisms by which their membrane binding properties are regulated are largely unknown. Many F-BAR proteins can be autoinhibited, usually through interactions between their C-terminal SH3 domain and N-terminal F-BAR/EFC domain (Guerrier et al, 2009; Rao et al, 2010; Guez-Haddad et al, 2015; Kelley et al, 2015; Stanishneva-Konovalova et al, 2016). Here, we suggest the proline-rich domain within the L1$_L$ of FBP17$_L$ and CIP4$_L$ may bind its own SH3 domain via the proline-rich binding RT loop (Saksela & Permi, 2012) and inhibit the C-terminal half of the protein. We demonstrate the F-BAR/EFC domain and the L1 region (FF$_S$—, CC$_S$—, FF$_L$—, and CC$_L$—) are sufficient to induce membrane tubulation in neurons and COS-7 cells. Our study suggests that binding of the SH3 domain to the PxxP motifs in the L1$_L$ may "close" the protein and inhibit binding of activated Rho GTPases such as Cdc42 and Rac1 to the HR1 domain. These data are consistent with a recent study showing that GTP-loaded Cdc42 is required to bring CIP4 and FBP17 to the membrane during early stages of fast endophilin-mediated endocytosis (Chan Wah Hak et al, 2018). Interestingly, pacsin/syndapin and Cdc15 have two PxxP motifs within analogous regions, suggesting these proteins may

also be inhibited if their SH3 domains are capable of binding these residues.

Our previous data suggest that the small GTPase Rac1, not Cdc42, is serving the role of recruiting CIP4$_S$ to the protruding plasma membrane (Saengsawang et al, 2013). Here, we show that wild-type or chimeric proteins that lack a long linker region and contain the CIP4 HR1 domain are able to localize to the periphery and inhibit filopodia (Fig 7B). Lacking the PxxP motifs may allow Rac1 to interact with the HR1 domain and actin-associated proteins to interact with the SH3 domain of CIP4, which may underlie the formation of actin ribs and veils (Saengsawang et al, 2013). This hypothesis is further supported by the CIP4$_L$-AxxA mutant, which localizes to the peripheral membrane in cortical neurons, likely because the HR1 domain and SH3 domains are accessible to binding partners. However, further work is required to determine the exact mechanism by which the activity of CIP4 family proteins is regulated.

# Materials and Methods

### Primary cortical neuronal cell culture

Swiss Webster mouse E14.5 cortical neurons are cultured in serum-free media which consists of Neurobasal (Invitrogen) with B27 supplement (Invitrogen), 2 mM glutamine (Invitrogen), 37.5 mM NaCl and 0.3% glucose at 37°C and 5% $CO_2$.

### COS-7 cell culture

COS-7 cells were cultured in DMEM (Gibco) with 10% FBS (HyClone) and 1% PenStrep (Gibco) at 37°C and 5% $CO_2$.

### HEK-293 cell culture

HEK-293 cells were cultured in DMEM, high-glucose (Gibco) with 10% FBS (HyClone), and 1% PenStrep (Gibco) at 37°C and 5% $CO_2$.

### Plasmids

Full-length CIP4$_S$-EGFP was a gift from Dr. Andrew Craig (Queen's University, Kingston, Ontario, Canada). CIP4$_S$-Tdtomato was previously made in our laboratory (Saengsawang et al, 2013), and full-length EGFP-FBP17$_L$ was a gift from Dr. Naoki Mochizuki (National Cardiovascular Center Research Institute, Osaka, Japan). An empty pCAX vector was a gift from Dr. Kate O'Connor-Giles (University of Wisconsin, Madison). EGFP was then cloned into the plasmid to create a pCAX-EGFP-N1 vector. All full-length proteins, chimeras, deletions, and mutations were cloned into this vector, resulting in C-terminally labeled proteins. There was no linker sequence between the final amino acid of the protein of interest and EGFP or mScarlet. These C-terminally labeled proteins were indistinguishable from proteins containing a longer linker region (GGGGSx3). Therefore, all of the plasmids used in this study did not have a linker sequence between the wild-type, chimeric, or deletion sequence and EGFP or mScarlet. Chimeras and deletions were constructed using gene Splicing by Overlap Extension (gene SOEing)

described previously (Horton et al, 2013). CIP4$_L$, CIP4$_L$-7Q, CIP4$_S$-7Q, FBP17$_L$-AxxA, and CIP4$_L$-AxxA were all created by cloning in long, double-stranded oligomers ordered from IDT into existing vectors using Gibson cloning. CA Cdc42 (Cdc42-V12) and Rac1 (Rac1-V12) and dominant negative Cdc42 (Cdc42-N17) were gifts from Dr. Timothy Gomez and have been verified in previous studies (Myers et al, 2012; Saengsawang et al, 2013). All constructs were verified by sequencing.

### Cortical neuron transfection

All mouse procedures were approved by the University of Wisconsin Committee on Animal Care and were in accordance with NIH guidelines. Cortical (E14.5) neuron cultures were prepared from Swiss Webster mice (Taconic) as described in previous publications (Viesselmann et al, 2011). Briefly, cortices were carefully dissected, trypsinized, and dissociated. Dissociated cortical neurons were resuspended in Nucleofector solution (Mouse Neuron Kit; Lonza) and transfected with an Amaxa Nucleofector II, according to the manufacturer's instructions. Transfected neurons were plated on 0.1 mg/ml poly-D-lysine (Sigma-Aldrich)–coated glass coverslips that were adhered to the bottom of 35-mm plastic culture dishes (the coverslip was placed over a 15-mm hole drilled through the bottom of the chamber). Neurons were suspended and plated in plating medium (Neurobasal medium with 5% FBS [HyClone], B27 supplement, 2 mM glutamine, 37.5 mM NaCl, and 0.3% glucose). After 1 h, the dishes were flooded with 2 ml serum-free medium, which is the plating medium without the FBS. Neurons were then imaged or fixed after 12 h in vitro (12 HIV).

### Immunocytochemistry

For fixed-cell imaging, cortical neurons and COS-7 cells were fixed in 4% paraformaldehyde/KREBs/sucrose at 37°C. Cultures were rinsed in PBS and blocked with 10% BSA/PBS, permeabilized in 0.2% Triton X-100/PBS, and labeled with phalloidin coupled to Alexa 568 (Invitrogen) to label actin filaments (1:50) and DAPI to label nuclei.

### Immunoblotting

HEK-293 cells were transfected at 70% confluency with 10 µg of each indicated plasmid using Lipofectamine 3000 (Invitrogen) following the manufacturer's protocol. Neurons were transfected with 5 µg of each indicated plasmid using Lonza's electroporation protocol and plated at a density of 2–3 million cells per well onto a poly-D-lysine treated six-well plate (0.1 mg/ml; Sigma-Aldrich). The cells were washed once with cold PBS before being lysed with 300 µl NP-40 lysis buffer (Invitrogen) with cOmplete Mini (Roche) and PhosStop (Roche) at 24 and 48 h post-transfection for HEK-293 and neurons, respectively. Lysate was spun at 21,000$g$ for 10 min, and supernatants were flash-frozen and stored at –80°C until use. The samples were thawed and loaded onto a 4–15% SDS Page gel (Bio-Rad), then transferred to PVDF membrane (Millipore). Membranes were blocked in 5% milk in TBS-T (0.1%), incubated with primary antibody overnight at 4°C and blotted with an HRP-containing secondary antibody for 1 h. TBS-T was used to wash the membrane, 3 × 15 min, after each incubation step. Antibodies used for

HEK-293 blots were goat-anti-GFP (1:1,000; Acris) and mouse anti-goat HRP (1:10,000; Jackson), and for neuron blots rabbit anti-HA (1:10,000; Abcam), mouse anti-tubulin (1:10,000; Sigma-Aldrich), goat-anti-GFP (1:1,000; Acris), and anti-goat and anti-mouse HRP secondary antibodies (1:10,000; Jackson) and HRP-Protein A (1:10,000; Invitrogen). Protein bands were visualized using Pierce ECL Western blotting substrate (Thermo Fisher Scientific).

### co-IP

Neurons and HEK-293 cells were grown, transfected and lysed following the above protocol, except IP Buffer was used to lyse cells (10% glycerol, 1% NP-40, 50 mM Tris, 200 mM NaCl, 2 mM MgCl$_2$, cOmplete Mini, and PhosStop). The lysates were thawed and loaded (250 μg for neurons and 500 μg for HEK-293 cells) onto Pierce Anti-HA magnetic beads (Thermo Fisher Scientific), and immunoprecipitated following the manufacturer's protocol. After the final wash, the beads were resuspended in 25 μl of 1× sample buffer (Invitrogen) and reducing buffer (Invitrogen), and then boiled for 5 min at 95°C. After boiling, everything (excluding the beads) was loaded onto the SDS–PAGE gel. Lysate lanes were loaded using 5% of total protein used for the IPs.

### Imaging

All fluorescence live-cell imaging was performed using a Nikon TIRF microscope as described (Hu et al, 2008). Briefly, the TIRF microscope consists of a Nikon TE2000E base with TIRF illuminator (Nikon), a Nikon 100×/1.49 NA Plan Apo TIRF objective, a Nikon perfect focus system for continuous automatic focusing of the sample during live imaging, a Nikon z-motor, a motorized x-y stage (Prior Scientific), a Lumen Pro200 fluorescent illumination system consisting of a 200 W metal halide lamp and a six-position excitation filter wheel and a fiber optic illuminator (Prior Scientific), a 10-position emission filter wheel, and a Coolsnap HQ-cooled interline CCD camera (Photometrics). For TIRF illumination, two lasers were used: a 40 mW argon laser for GFP illumination and a 10-mW solid-state 561-nM laser for DsRed2, mCherry, Tdtomato, and mRuby illumination (both Melles-Griot). The microscope was equipped with a dual wavelength (EGFP/mCherry) dichroic mirror (z488/561rdc, Chroma) for both TIRF and wide-field illumination. This system allowed us to collect two-color TIRF images. During live imaging, neurons were kept at 37°C and the culture dish was closed with a glass ring, coverslip, and silicone grease. All images were binned 2 × 2 and collected, measured, and analyzed in FIJI imaging software. Pearson's correlation coefficient was calculated using FIJI software with the Coloc2 plugin (Bolte & Cordelieres, 2006), where a coefficient close to 1.0 indicates complete colocalization, and 0 indicates no colocalization. Figures were compiled in Photoshop and Illustrator (Adobe). Fixed COS-7 cell imaging was performed using a Zeiss confocal microscope. The confocal microscope consists of a Zeiss LSM800 base with Airyscan; a Zeiss 63×/1.4 NA plan Apochromat objective; a Zeiss scanning stage with stepper motor; 405-, 488-, 561-, and 633-nm laser lines; and Zen 2.3 imaging software. For drug studies, the Cdc42 inhibitor ZCL278 (Tocris) and the Rac1 inhibitor NSC23766 (Tocris) were used at final concentrations of 50 and 100 μM, respectively. They were added to cultures, incubated for 15 min, and then washed out for 10 min.

### Image analysis

Images acquired from the TIRF system described above were 696 × 520 pixels, 16-bit. Displayed images were generated with FIJI software. All graphs and plots were generated in Prism7 (GraphPad). Peripheral intensity measurements were measured by taking the average ratio of the average intensity of four spots (with a radius of 0.15 μm) on the perimeter of the cell to four spots 3 μm inside the perimeter (each data point represents the average of four ratios). Filopodia were quantified as any point that protrudes from the cell any more than 1 μm. The perimeter for filopodia measurement in stage 1 neurons was determined by manually drawing a polygon encompassing most of the cell but excluding protrusions from the main cell body. Tubules were quantified as any elongated shape, which has a length that was three times its width, excluding the periphery. Quantification for complexity was determined by thresholding the mRuby-Lifeact image and tracing the cell perimeter, this trace was then overlaid onto the EGFP image to ensure the trace was accurate. The complexity measure is showing the ratio of the cell perimeter to the area of the cell and offers a description of cell morphology. Colocalization was determined by drawing an ROI that completely encompasses the cell and using the Coloc2 FIJI plugin.

### Statistics

P values were determined by one-way ANOVA with Kruskal–Wallis post-test multiple comparisons and noted in the figure legends. The Kruskal–Wallis post-test multiple comparisons were selected for our analysis as most of our data did not show a normal distribution, and this post-test has more stringent requirements for significance. Most graphs are shown as a box-and-whisker plot where the box extends from the 25th to 75th percentiles and the line in the middle of the box represents the median. Whiskers are drawn down to the 10th percentile and up to the 90th, and points below and above the whiskers are drawn as individual points. The number of cells (n) is noted in the figure legends. For each data set, cells from at least three independent experiments were quantified. Significance was denoted as follows: *$P < 0.05$, **$P < 0.01$, ***$P < 0.001$, and ****$P < 0.0001$.

## Supplementary Information

## Acknowledgements

We thank all the members of the Dent lab for their helpful discussions and critical comments on the manuscript. We thank Sayantanee Biswas for the help with plasmid design and cloning. We thank Witchuda Saengsawang for necessary input and help throughout the project. We thank the Gomez,

Huang, and Kalil labs for their generosity in sharing equipment, reagents, and knowledge and the developmental neurobiology group at UW for critical advice during the project. This work was supported by grants from NIH to EW Dent (R01NS080928) and KL Taylor (T32GM07507) and the University of Wisconsin Vilas Associates Award to EW Dent.

## Author Contributions

K Taylor: conceptualization, data curation, formal analysis, supervision, validation, investigation, visualization, methodology, and writing—original draft, review, and editing.
R Taylor: conceptualization, data curation, formal analysis, investigation, and writing—review and editing.
K Richters: conceptualization, investigation, methodology, and writing—review and editing.
B Huynh: investigation.
J Carrington: investigation.
M McDermott: investigation.
R Wilson: investigation.
EW Dent: conceptualization, data curation, supervision, funding acquisition, visualization, methodology, project administration, and writing—original draft, review, and editing.

## Conflict of Interest Statement

The authors declare that they have no conflict of interest.

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
