## [Reviewer comments · Life Science Alliance]

Life Science Alliance

Opposing functions of F-BAR proteins in membrane protrusion, tubule formation and neurite outgrowth

Kendra Taylor, Russell Taylor, Karl Richters, Brandon Huynh, Justin Carrington, Maeve McDermott, Rebecca Wilson, and Erik Dent

DOI: <https://doi.org/10.26508/lsa.201800288>

Corresponding author(s): Erik Dent, University of Wisconsin-Madison

Review Timeline:

Submission Date:	2018-12-20
Editorial Decision:	2019-01-28
Revision Received:	2019-04-12
Editorial Decision:	2019-05-06
Revision Received:	2019-05-15
Accepted:	2019-05-17

Scientific Editor: Andrea Leibfried

Transaction Report:

January 28, 2019

Re: Life Science Alliance manuscript #LSA-2018-00288-T

Prof. Erik Dent
University of Wisconsin-Madison
Department of Anatomy
Department of Neuroscience 5431 WIMR2 1111 Highland Ave.
1111 Highland Ave.
Madison, WI 53705-2275

Dear Dr. Dent,

Thank you for submitting your manuscript entitled "Opposing functions of F-BAR proteins in membrane protrusion, tubule formation and neurite outgrowth" to Life Science Alliance. The manuscript was assessed by expert reviewers, whose comments are appended to this letter.

As you will see, the reviewers all point out that the technical quality is overall good though a few controls are missing. However, they also think that the overall value provided to the field is quite limited and that the overexpression system used impedes advancing our understanding of biological function. Furthermore, they would have expected inclusion of an analysis on the interplay of Cip4 and FBP17 with Rho GTPases.

Given this input, we concluded that we could invite you to submit a revised version of your work, should you be prepared to not only addressing the technical issues noted (lack of controls noted by reviewer #1 as well as reviewer #2, point 3), but to also extending the work along the lines suggested by the reviewers. We would need strong support from the reviewers on such a revised version for publication here.

The typical timeframe for revisions is three months. Please note that papers are generally considered through only one revision cycle.

Thank you for this interesting contribution to Life Science Alliance. We are looking forward to receiving your revised manuscript.

Sincerely,

- A letter addressing the reviewers' comments point by point.
- An editable version of the final text (.DOC or .DOCX) is needed for copyediting (no PDFs).
- High-resolution figure, supplementary figure and video files uploaded as individual files: See our detailed guidelines for preparing your production-ready images, <http://life-science-alliance.org/authorguide>
- Summary blurb (enter in submission system): A short text summarizing in a single sentence the study (max. 200 characters including spaces). This text is used in conjunction with the titles of papers, hence should be informative and complementary to the title and running title. It should describe the context and significance of the findings for a general readership; it should be written in the present tense and refer to the work in the third person. Author names should not be mentioned.

B. MANUSCRIPT ORGANIZATION AND FORMATTING:

Full guidelines are available on our Instructions for Authors page, <http://life-science-alliance.org/authorguide>

Reviewer #1 (Comments to the Authors (Required)):

Cip4 and FBP17 are two related F-BAR proteins that are able to bind membrane phospholipids, initiate membrane curvature and scission via Src homology-3 (SH3) domain interaction with their diverse binding partners, thus regulating fundamental cellular functions, including endocytosis, phagocytosis, filopodial/lamellipodial protrusions, cytokinesis, adhesion, and podosome formation, via distinct signaling pathways determined by specific domain-binding partners.

In the present manuscript "Opposing functions of F-BAR proteins in neuronal membrane protrusion, tubule formation and neurite outgrowth", Taylor et al. follow up on their previous studies on Cip4 and FBP17 localization and filopodia formation (please see Saengsawang, et al., 2012 and Saengsawang, et al., 2013). Again, the authors used an in vitro cell culture approach/strategy using primary cortical neurons transfected with different EGFP-tagged Cip4- and FBP17-constructs. By overexpressing distinct EGFP-tagged deletion and swapping constructs they identified an important role of the first linker region (L1) of Cip4 and FBP17 in defining localization and filopodia formation activity. They further conclude that the L1S linker together with the Cdc42-interacting HR1 domain determines lamellipodial localization, whereas neurite (filopodial) outgrowth inhibition depends on the SH3 domain.

The paper is a diligent piece of work with numerous main and supplemental figures (13!), overall well written, and most data are well documented and described. However, this work is too descriptive and the physiological relevance of the gain-of-function in vitro data remains unclear. The overexpression data suggest a differential domain requirement for localization and neurite morphology Cip4 and FBP17, but without additional functional data the overall findings of this manuscript are quite limited. Here, a more sophisticated structure-function analysis in mutant background would be required that allows further novel insights into the in vivo function and regulation of both similar proteins, beyond their previous published papers in 2012 and 2013. Additional functional experiments should also address important questions, such as what makes the differences in the localization and gof phenotypes in neuronal and epithelial/non-neuronal cell types? Are there distinct binding partners that differentially bind to first linker region (L1), HR1 or the SH3 domains, and thereby their activity? If Rac1, but not Cdc42, „is serving the role of recruiting CIP4S to the protruding plasma membrane", what is the function of the HR1 domain in FBP17? Members of the Cip4/Toca-1 F-BAR subfamily are also known inhibitors of Diaphanous-related formins. Do formins mediate FBP17-induced neurite outgrowth?

Minor points:

- 1) Co-immunoprecipitation experiments lack essential controls (e.g. IP with preimmune sera).
- 2) The authors found that expression of the F-BAR/EFC (1-300aa) domain of either CIP4 or FBP17 was not sufficient to localize these proteins to membranous structures in either neurons or COS-7 cells. However, this interesting observation was already made 10 years ago. What makes the difference between membrane binding in vitro and in vivo?
- 3) For statistics, quantifications were often done on samples consisting of very small cell numbers (22-30 cells), and should be increased. Most images only showed single cells? What is the transfection efficiency of these primary neurons?

Reviewer #2 (Comments to the Authors (Required)):

Comments for Taylor et al.

This paper contains significant effort to describe the different localization of CIP4 to FBP17 in neuronal cells, not in the other cultured cells. They examined the localization of a lot of domain swapping mutants and found the important region for the protein localizations. The overall study appeared to be performed under good technical quality. However, I think the reason for the differential localization in neurons are not clear enough. I would like to point several issues to improve the manuscript.

1. The localization difference of CIP4 to FBP17 was observed only in the neuronal cells, and therefore it would be reasonable to think the binding proteins to these proteins are different dependent on the cell types. Then, FBP17/Rapostlin was shown to bind to the small GTPase Rnd2 (The Journal of Biological Chemistry 277, 45428-45434), while CIP4 binds to Cdc42. The difference in the preference to the small GTPases appeared to be another reasonable reason. The HR1 domain swapping also resulted in the differential localizations, supporting this idea (Figure 2).
2. To test the small GTPase-dependency, it might be better to examine the role of small GTPases by the expression of the dominant negative mutants or siRNA-induced reduction of the proteins.
2. I agree the poly-basic region and the proline-rich region contribute to the localizations of FBP17 and of CIP4. However, these regions are conserved between these two proteins, and therefore appeared to be unlikely to be the reason for differential localizations. The sequence comparison of FBP17 to CIP4 should be included in Figure 6A.
3. The expression levels of overexpressed proteins should be shown by western blot, in comparison with the endogenous protein expression levels.
4. The detailed construct information including the linker sequences from EGFP to CIP4/FBP17 is helpful, because the difference in the linker might induce the difference in the localizations. The domain swapped constructs are a little bit confusing, and I prefer to put the alignment of all of the amino-acid sequences of the constructs they used.

Reviewer #3 (Comments to the Authors (Required)):

First of all, I would like to mention that I liked reading the paper by Taylor et al., and I find the conclusions to be sound. The article deals with the unique and opposing function of the two F-BAR domain-containing proteins CIP4 and FBP17. There are several splice variants of each of these two proteins but of specific interest for neuronal cells are the short splice variant of CIP4 (CIP4S) and the long splice form of FBP17 (FBP17L). The authors have made a major effort to characterize the domains and motifs responsible for the respective phenotype of CIP4S and FBP17L. I find the data highly convincing. I think the figures are sufficient as they are.

Having this said, there is one parameter that I am lacking and that I think the authors should analyze, that is the interaction to Rho GTPases. CIP4 was originally identified in a yeast two-hybrid screen for Cdc42-interacting proteins (hence its rather boring name). The original article (Aspenström, Curr Biol., 1997) indicated that Cdc42 induced CIP4 localization to the cell periphery and to cellular structures that in retrospect must have been membrane tubules, which is in contrast to what the authors claim in the last paragraph of the Discussion. They rather put forward Rac1 as responsible for membrane targeting of CIP4 but I cannot find the evidences for this notion. There are also conflicting views on if FBP17 actually binds Rho GTPases at all. Therefore, I think it would be of significance to examine the interaction between the CIP4/FBP17 chimeras and Cdc42 and to see if any of the chimeras can be specifically targeted to the cell periphery/membrane tubules in response to Cdc42 (and Rac1-since the authors propose that CIP4 is targeted to the cell periphery

in a Rac1-dependent manner). I do not request the authors to test all the mutants, just the key chimeras.

Reviewer #1

....In the present manuscript "Opposing functions of F-BAR proteins in neuronal membrane protrusion, tubule formation and neurite outgrowth", Taylor et al. follow up on their previous studies on Cip4 and FBP17 localization and filopodia formation (please see Saengsawang, et al., 2012 and Saengsawang, et al., 2013).

The reviewer is correct that we have studied CIP4 in these two publications, but we have never published on FBP17. There are no FBP17 data in either of those two publications.

Again, the authors used an in vitro cell culture approach/strategy using primary cortical neurons transfected with different EGFP-tagged Cip4- and FBP17-constructs. By overexpressing distinct EGFP-tagged deletion and swaping constructs they identified an important role of the first linker region (L1) of Cip4 and FBP17 in defining localization and filopodia formation activity. They further conclude that the L1S linker together with the Cdc42-interacting HR1 domain determines lamellipodial localization, whereas neurite (filopodial) outgrowth inhibition depends on the SH3 domain. The paper is a diligent piece of work with numerous main and supplemental figures (13!), overall well written, and most data are well documented and described.

Thank you for appreciating the amount of work that went into this manuscript.

However, this work is too descriptive and the physiological relevance of the gain-of-function in vitro data remains unclear. The overexpression data suggest a differential domain requirement for localization and neurite morphology Cip4 and FBP17, but without additional functional data the overall findings of this manuscript are quite limited. Here, a more sophisticated structure-function analysis in mutant background would be required that allows further novel insights into the in vivo function and regulation of both similar proteins, beyond their previous published papers in 2012 and 2013.

We believe this work goes well beyond our 2012 and 2013 papers. The 2012 and 2013 publications do not contain any FBP17 data, thus all FBP17 data presented here is novel. Indeed, this the first publication to show any role for FBP17 in neurite outgrowth.

We are not sure what the reviewer means by a "more sophisticated structure-function analysis in a mutant background". Although many studies use point and deletion mutants to determine the function of proteins, few studies swap protein domains to make chimeric proteins. We made 20+ chimeric proteins, in addition to deletion mutants, and quantified their effects with four different protein localization and morphological measurements, as well as measuring neurite and axon outgrowth through three stages of development in living neurons.

Regarding a mutant background, we confirmed that full length proteins (Fig. S1K) and a few of the chimeras (L1 swap (Fig. S4B) and data not shown) act exactly the same in wild-type and CIP4 KO neurons. We also provide reasoning in the results why we cannot use knockdown of FBP17 (see second paragraph on page 6 of Results). If the reviewer is suggesting that we conduct more experiments in the mouse, we believe that is a request that goes well beyond the scope of this paper.

Additional functional experiments should also address important questions, such as what makes the differences in the localization and gof phenotypes in neuronal and epithelial/non-neuronal cell types?

We believe this is an important question, but is beyond the scope of this study. Our present hypothesis is that differential localization of CIP4 in neurons and non-neuronal cells may be due to: (1) differential expression of specific proteins in neurons and non-neuronal cells, (2) differential activation of GTPases in these two cell types and/or (3) differential post-translational modifications of F-BAR proteins in these two cell types. We are actively pursuing these avenues in a separate study and believe it is likely to be a combination of all of these possibilities. Nevertheless, we have added a figure (Fig. S8A) showing that CA-Rac1 expression is sufficient to cause CIP4_L to relocate from tubules to the peripheral protruding membrane in neurons. However, the same treatment in COS-7 cells does not result in CIP4_L re-localizing to the periphery (Fig. S8B). Thus, activation of Rac1 plays an important role in targeting CIP4 to the periphery in neurons but Rac1 is not sufficient to localize CIP4 to the periphery in COS-7 cells.

Are there distinct binding partners that differentially bind to first linker region (L1), HR1 or the SH3 domains, and thereby their activity?

We do not know if the CIP4 or FBP17 L1 regions bind different partners but they seem indistinguishable when we swap them. In Figure 5C-H we show that swapping the short FBP17 linker into CIP4_S has no effect on any of our measures. We address the HR1 domain below. We have not directly tested if there are different binding partners of the SH3 domain of CIP4 and FBP17, but it is well established in the literature that they are known to bind different, but overlapping, sets of proteins. However, we find very little difference when we swap the SH3 domain of CIP4 and FBP17 (Fig. S2A-G).

If Rac1, but not Cdc42, „is serving the role of recruiting CIP4S to the protruding plasma membrane“, what is the function of the HR1 domain in FBP17?

We have added data that Cdc42 affects FBP17_L, potentially through the HR1 domain. This figure shows inhibition of Cdc42, either through expression of dominant negative Cdc42 (Fig. S3A, B) or inhibition of Cdc42 pharmacologically (Fig. S3C, D), reduces the number of FBP17_L-containing tubules. However, the pharmacological inhibition of Rac1 does not have any effect on the number of FBP17_L tubules (Fig. S3E). Thus, the HR1 domain of FBP17 is likely functioning with Cdc42, either directly or indirectly. These results are exactly opposite of what we previously showed for CIP4 (Saengsawang et al., 2013), where Rac1, but not Cdc42, affects localization and function of CIP4.

Members of the Cip4/Toca-1 F-BAR subfamily are also known inhibitors of Diaphanous-related formins. Do formins mediate FBP17-induced neurite outgrowth?

This is an interesting question and we hope to determine if these filopodia are Diaphanous-related formin- or Ena/VASP-dependent in the future. We showed previously that both the Diaphanous-related formin DAAM1 and the actin polymerase Ena/VASP are associated with CIP4-based protrusions (Saengsawang et al., 2013) and we know from another previous study (Dent et al., 2007) that embryonic cortical neurons do not contain mDia2. Thus, we hope to conduct future studies that will test both of these proteins, and possibly others, in FBP17-induced filopodia. However, we feel this question does not directly relate to the conclusions of the study.

Minor points:

1) Co-immunoprecipitation experiments lack essential controls (e.g. IP with preimmune sera).

We used magnetic beads coated with anti-HA antibodies. Beads coupled to preimmune sera are not available. To show specificity of our blots we have now included the GFP lane as a negative control to show it does not IP with the anti-HA beads (Fig. 1K). We also reprobbed the blot with an anti-HA antibody to show CIP4 is expressed in all lysate lanes and it comes down in all IP lanes (positive control). Furthermore, we reprobbed the blot a second time for tubulin to show it is in all of the lysate lanes but does not IP with anti-HA (a second negative control). In this new blot we present there is less FBP17_L, compared to CIP4_S, in the lysate because FBP17_L does not express as well as CIP4_S in these cells. Nevertheless, we never found FBP17_L coming down with CIP4_S in three separate experiments and in Fig. 1L we normalized the amount of pulldown to the levels of input in each experiment. These new data confirm that CIP4_S is not immunoprecipitating with FBP17_L.

2) The authors found that expression of the F-BAR/EFC (1-300aa) domain of either CIP4 or FBP17 was not sufficient to localize these proteins to membranous structures in either neurons or COS-7 cells. However, this interesting observation was already made 10 years ago. What makes the difference between membrane binding in vitro and in vivo?

We apologize for not including the reference to Fricke et al., 2009, where this observation, to which the reviewer is referring, was made. This paper showed the F-BAR region of *Drosophila* CIP4 (dCIP4/Toca-1) was able to induce membrane tubules *in vitro* but did not appear to form tubules in *Drosophila* S2 cells (although it did concentrate on unknown elongated structures in the cytoplasm of these cells). We have now included this reference in the discussion and indicate that our results are consistent with this observation (although when we expressed the F-BAR alone it was simply diffuse in both cell types, rather than forming unknown elongated structures, as it did in S2 cells). Since dCIP4 does not appear to have a poly-basic region after the F-BAR/EFC domain (see amino acid sequence alignment in Fig. S7A) the data in Fricke et al., are consistent with our data showing that the poly-basic region is necessary for membrane binding and bending in mammalian cells. We suspect that the membrane structure in cells is more complex than liposomes with regard to both the phospholipids and proteins that are present. Thus, the poly-basic region in CIP4 and FBP17 may be required to bind and bend this more complex form of membrane.

3) For statistics, quantifications were often done on samples consisting of very small cell numbers (22-30 cells), and should be increased. Most images only showed single cells? What is the transfection efficiency of these primary neurons?

Our transfection efficiency is in the neighborhood of 20-40%. However, we specifically plated the primary neurons at low density so that they were not in contact with other cells, which would impede our ability to take measurements on stage 1 neurons, as well as compromise our measurement of stage progression (neurite and axon outgrowth). As far as cell numbers, all of the data shown in the paper were quantified from cultures of cortical neurons from at least three separate litters of mice in which we took a random sampling of approximately seven to ten living neurons from each preparation. For quantifying neuron maturation (stage 1-3) our numbers were higher (50-80 neurons from at least three separate experiments). These numbers were based upon previous studies we have conducted (Dent et al., 2007; Saengsawang et al., 2012, 2013). In all of the graphs we have highly significant differences ($p < 0.01$ to $p < 0.0001$) between certain conditions and insignificant differences between others, indicating that the statistical power of the experiments is appropriate with the number of neurons used.

Reviewer #2

Comments for Taylor et al.

This paper contains significant effort to describe the different localization of CIP4 to FBP17 in neuronal cells, not in the other cultured cells. They examined the localization of a lot of domain swapping mutants and found the important region for the protein localizations. The overall study appeared to be performed under good technical quality. However, I think the reason for the differential localization in neurons are not clear enough. I would like to point several issues to improve the manuscript.

1. The localization difference of CIP4 to FBP17 was observed only in the neuronal cells, and therefore it would be reasonable to think the binding proteins to these proteins are different dependent on the cell types. Then, FBP17/Rapostlin was shown to bind to the small GTPase Rnd2 (The Journal of Biological Chemistry 277, 45428-45434), while CIP4 binds to Cdc42. The difference in the preference to the small GTPases appeared to be another reasonable reason. The HR1 domain swapping also resulted in the differential localizations, supporting this idea (Figure 2).

These are logical hypotheses that we are pursuing in a separate study (see response to reviewer #1). We are in the process of performing immunoprecipitations, followed by mass spectrometry to determine the set of proteins that interact with CIP4 in neurons vs. COS-7 cells (FBP17 shows tubule localization in both cell types). With regard to small GTPases, we have published that CIP4 localization to the periphery of cortical neurons depends on Rac1, rather than Cdc42 (Saengsawang et al., 2013 J Cell Science, 126: 2411-2423). Cdc42 appears to be the primary GTPase associated with CIP4 in non-neuronal cells (several studies), where CIP4 localizes to tubules. Thus, we favor the idea of associations with different GTPases in different cell types, but to thoroughly determine the answer to this question is beyond the scope of this paper.

Regarding the assertion that the HR1 domain swapping resulted in differential localizations (Fig. 2), we do believe that the CIP4 HR1 domain is distinct from the FBP17 HR1 domain. Indeed, in Fig. 5 we show that FFsC-- has a peripheral localization, while FFsF-- has a tubule localization (the only difference being the HR1 domains). Moreover, we have included two new figures (Figs. S3 and S8) showing that Rac1 activity is sufficient to localize CIP4L (normally on

tubules) to the peripheral protruding membrane in neurons, but not in COS-7 cells. We also present data showing that FBP17_L is dependent on Cdc42 (Fig. S3A-D) but not Rac1 (Fig. S3E and Fig. S8A). Thus, we favor the interpretation that the HR1 domain of CIP4 is associating with Rac1, while FBP17 is associating with Cdc42 in neurons. In COS-7 cells (Fig. S8B) CIP4 may be slightly affected by activating Rac1, but it is not sufficient to localize CIP4L from tubules to peripheral protruding membrane.

2. To test the small GTPase-dependency, it might be better to examine the role of small GTPases by the expression of the dominant negative mutants or siRNA-induced reduction of the proteins.

We have used dominant negative mutants and constitutively active mutants of Cdc42, Rac1 and RhoA, as well as pharmacological compounds, in our previous study (Saengsawang et al., 2013). This study showed that the peripheral localization of CIP4 in neurons is increased with CA-Rac1 and decreased with CA-Cdc42, while CA-RhoA had no effect on CIP4 peripheral localization. We have also included two new figures as mentioned in point 1 above.

2. I agree the poly-basic region and the proline-rich region contribute to the localizations of FBP17 and of CIP4. However, these regions are conserved between these two proteins, and therefore appeared to be unlikely to be the reason for differential localizations. The sequence comparison of FBP17 to CIP4 should be included in Figure 6A.

As we describe in the model in Figure 7, the reason for the different localizations of CIP4 and FBP17 in neurons is because only the short form of CIP4 (CIP4_S) and the long form of FBP17 (FBP17_L) are expressed in neurons. The short form of CIP4, which does not contain the PxxP region, but does contain the CIP4 HR1 domain, localizes to the peripheral protruding membrane. However, the long form of FBP17 (FBP17_L) localizes to tubules, as does the long form of CIP4 (CIP4_L), when they are exogenously expressed in neurons.

Instead of adding the polybasic sequence comparisons in Fig. 6A, which focuses exclusively on CIP4, we have added the sequence comparisons of the polybasic region for FBP17 and CIP4 in Fig. 6G. The sequence comparison for both the polybasic and poly-PxxP regions are in Supplemental Figure 7A as well.

3. The expression levels of overexpressed proteins should be shown by western blot, in comparison with the endogenous protein expression levels.

It is very difficult to directly compare levels of overexpressed protein in neurons because, (1) only a percentage of cells are transfected (~30%, but this varies from prep to prep) and (2) there is not an available antibody to CIP4 that is specific for immunocytochemistry (all antibodies label knockout tissue and dissociated cells – which we have mentioned in a previous publication – Saengsawang et al., 2012). Thus, we cannot compare brightness between transfected and untransfected individual cells. Importantly, the cells we chose to image and quantify have low to medium levels of expression of GFP/RFP fluorescence, compared to other transfected neurons in the dish. We have added this information in the Results section.

Additionally, we have very recently generated mouse embryonic stem cells with endogenously labeled CIP4 (via CRISPR/Cas9 editing), which can be differentiated into forebrain neurons via treatment with EGF, FGF and Heparin to generate neuronal progenitor cells, and then subsequent removal of these factors to generate neurons. This endogenously labeled CIP4 localizes to the protruding periphery in stage 1 lamellar neurons (see image below – for reviewers only). Thus, we don't believe that overexpressing CIP4 at low-to-medium levels in mouse cortical neurons results in overexpression artifacts.

4. The detailed construct information including the linker sequences from EGFP to CIP4/FBP17 is helpful, because the difference in the linker might induce the difference in the localizations. The domain swapped constructs are a little bit confusing, and I prefer to put the alignment of all of the amino-acid sequences of the constructs they used.

All of our constructs are C-terminally labeled with no linker sequence in identical pCAX plasmids (as we now outline in the Methods). Our previous work indicated that labeling this end of the protein, compared to the N-terminus, resulted in less labeled protein in the cytoplasm (presumably non-functional) and more protein either at the periphery or on tubules. The fluorescent protein (either EGFP or mScarlet) is fused just after the final amino acids of the protein, with no linker. We used this same methodology in our previous two publications (Saengsawang et al., 2012; 2013). Recently, we have made a construct with a longer linker (GGGSx3) between the protein and EGFP. This construct results in an indistinguishable phenotype to the fusion protein without the linker. Thus, for the present study we have used plasmids with no linker between the protein of interest and EGFP or mScarlet to be consistent with our previous studies. We have added these data to the Methods section.

Reviewer #3

First of all, I would like to mention that I liked reading the paper by Taylor et al., and I find the conclusions to be sound.... The authors have made a major effort to characterize the domains and motifs responsible for the respective phenotype of CIP4S and FBP17L. I find the data highly convincing. I think the figures are sufficient as they are.

Thank you. We appreciate the acknowledgement of our efforts.

Having this said, there is one parameter that I am lacking and that I think the authors should analyze, that is the interaction to Rho GTPases. CIP4 was originally identified in a yeast two-hybrid screen for Cdc42-interacting proteins (hence its rather boring name). The original article (Aspenström, Curr Biol., 1997) indicated that Cdc42 induced CIP4 localization to the cell periphery and to cellular structures that in retrospect must have been membrane tubules, which is in contrast to what the authors claim in the last paragraph of the Discussion. They rather put forward Rac1 as responsible for membrane targeting of CIP4 but I cannot find the evidences for this notion.

The evidence that Rac1, instead of Cdc42, is responsible for membrane targeting of CIP4 to the cell periphery in neurons is in Figure 2 of our previous paper (Saengsawang et al., 2013 J Cell Science, 126: 2411-2423). In that figure we show, using both DN-Rac1 and CA-Rac1, as well as the Rac inhibitory compound NSC23766, that CIP4 localization at the protruding edge of stage 1 neurons is due to Rac1 activity. Furthermore, we show that DN-Cdc42 increases the concentration of CIP4 at the periphery. We present a model at the end of the Saengsawang (2013) paper showing that membrane targeting of CIP4 to the periphery is associated with PIP3, WAVE, Rac1, Mena and DAAM1. We have also included an additional figure in this manuscript that shows that CIP4_L, which normally is localized to tubules, localizes to the periphery if expressed with CA-Rac1 in neurons, but not COS-7 cells (Fig. S8). These data indicate that active Rac1 is responsible for localizing CIP4 to the periphery in neurons.

There are also conflicting views on if FBP17 actually binds Rho GTPases at all. Therefore, I think it would be of significance to examine the interaction between the CIP4/FBP17 chimeras and Cdc42 and to see if any of the chimeras can be specifically targeted to the cell periphery/membrane tubules in response to Cdc42 (and Rac1-since the authors propose that CIP4 is targeted to the cell periphery in a Rac1-dependent manner). I do not request the authors to test all the mutants, just the key chimeras.

We have added another figure (Figure S3) showing that the localization of FBP17 to membrane tubules requires active Cdc42. Using both DN-Cdc42 (Fig. S3A, B) and a drug that inhibits Cdc42 (ZCL278) (Fig. S3D) we show that the number of FBP17 tubules decreases, while inhibition of Rac1 with NSC23766 has no effect on FBP17 tubule number (Fig. S3E). However, CA-Cdc42 did not increase the number of tubules (Fig. S3B). Although tubule number decreases with DN-Cdc42 or pharmacological inhibition of Cdc42, we never see FBP17 go to the periphery when Cdc42 is activated (Fig. S3A). Even when we include CA-Rac1, FBP17_L does not go to the periphery (Fig. S8A), indicating it is not responsive to Rac1 activity. Moreover, we have shown in our previous study (Saengsawang et al., 2013) that activation of Rac1 increases levels of CIP4_S at the cell periphery, while activation of Cdc42 decreases levels of CIP4_S at the periphery.

May 6, 2019

RE: Life Science Alliance Manuscript #LSA-2018-00288-TR

Prof. Erik W Dent
University of Wisconsin-Madison
Department of Neuroscience
5431 WIMR2
1111 Highland Ave.
Madison, WI 53705-2275

Dear Dr. Dent,

Thank you for submitting your revised manuscript entitled "Opposing functions of F-BAR proteins in membrane protrusion, tubule formation and neurite outgrowth". As you will see, the reviewers appreciate the introduced changes and we would thus be happy to publish your paper in Life Science Alliance pending final minor revisions:

- please address reviewer #2's remaining comments
- please upload all figures (also suppl figures) as individual files and the manuscript text as a docx file
- please add the p-values depicted to the figure legends - I appreciate that you provide the necessary information in the material & methods section, but think that it would be useful for readers to find the information in the legends as well

A. FINAL FILES:

- An editable version of the final text (.DOC or .DOCX) is needed for copyediting (no PDFs).
- High-resolution figure, supplementary figure and video files uploaded as individual files: See our detailed guidelines for preparing your production-ready images, <http://www.life-science-alliance.org/authors>
- Summary blurb (enter in submission system): A short text summarizing in a single sentence the

study (max. 200 characters including spaces). This text is used in conjunction with the titles of papers, hence should be informative and complementary to the title. It should describe the context and significance of the findings for a general readership; it should be written in the present tense and refer to the work in the third person. Author names should not be mentioned.

B. MANUSCRIPT ORGANIZATION AND FORMATTING:

Sincerely,

Andrea Leibfried, PhD
Executive Editor
Life Science Alliance
Meyrhofstr. 1
69117 Heidelberg, Germany
t +49 6221 8891 502
e a.leibfried@life-science-alliance.org
www.life-science-alliance.org

Reviewer #1 (Comments to the Authors (Required)):

Overall, I am satisfied with the improved version of the manuscript. The authors have addressed my concerns sufficiently to recommend publication.

Reviewer #2 (Comments to the Authors (Required)):

I Comments for Taylor et al.

The manuscript appeared to be significantly improved, but sometimes the results and the role of each region of the FBP17 or CIP4 are difficult to be followed. There is a summary cartoon, but also I would suggest to put the result table showing the localization at the periphery and the filopodia as well as the phenotyping (cell staging) dependent on the each region of the chimeric protein.

Reviewer #3 (Comments to the Authors (Required)):

My summary and comments were included in my previous report, it is not necessary to repeat my statements. The authors have satisfactorily responded to my questions (and the questions by the other reviewers as far as I can judge). I recommend the article to be published.

May 17, 2019

RE: Life Science Alliance Manuscript #LSA-2018-00288-TRR

Prof. Erik W Dent
University of Wisconsin-Madison
Department of Neuroscience
5431 WIMR2
1111 Highland Ave.
Madison, WI 53705-2275

Dear Dr. Dent,

Thank you for submitting your Research Article entitled "Opposing functions of F-BAR proteins in membrane protrusion, tubule formation and neurite outgrowth". It is a pleasure to let you know that your manuscript is now accepted for publication in Life Science Alliance. Congratulations on this interesting work.

DISTRIBUTION OF MATERIALS:

Again, congratulations on a very nice paper. I hope you found the review process to be constructive and are pleased with how the manuscript was handled editorially. We look forward to future exciting submissions from your lab.

Sincerely,
